# Electrochemical on-surface synthesis of a strong electron-donating graphene nanoribbon catalyst

Hiroshi Sakaguchi [1] ✉, Takahiro Kojima [1], Yingbo Cheng[1], Shunpei Nobusue [1] & Kazuhiro Fukami [2]

On-surface synthesis of edge-functionalized graphene nanoribbons (GNRs) has attracted much attention. However, producing such GNRs on a large scale through on-surface synthesis under ultra-high vacuum on thermally activated metal surfaces has been challenging. This is mainly due to the decomposition of functional groups at temperatures of 300 to 500 °C and limited monolayer GNR growth based on the metal catalysis. To overcome these obstacles, we developed an on-surface electrochemical technique that utilizes redox reactions of asymmetric precursors at an electric double layer where a strong electric field is confined to the liquid-solid interface. We successfully demonstrate layer-by-layer growth of strong electron-donating GNRs on electrodes at temperatures <80 °C without decomposing functional groups. We show that high-voltage facilitates previously unknown heterochiral di-cationic polymerization. Electrochemically produced GNRs exhibiting one of the strongest electron-donating properties known, enable extraordinary silicon-etching catalytic activity, exceeding those of noble metals, with superior photoconductive properties. Our technique advances the possibility of producing various edge-functional GNRs.

Graphene nanoribbons (GNRs), one-dimensional graphene structures, have gained significant attention for their electronic properties, which can be controlled by altering their widths and edges[1-3]. On-surface synthetic techniques on thermally activated metal surfaces under ultrahigh vacuum (UHV) have been used to create various hydrocarbon-based GNRs through precursor polymerization followed by dehydrogenation[4-14]. On-surface synthesis can provide high purity and organization due to the gas-solid interface, making it advantageous over solution-based organic synthesis, except for the difficulty of large-scale production[15]. Edge functionalization of GNRs by introducing electron-donating or withdrawing substituents into their backbones is crucial for their practical use in electronic, magnetic, and energy applications[16-21]. However, on-surface synthesis for edge-functionalization is challenging due to the thermal degradation of fragile substituents at 300 to 500 °C[22-24], except for remarkably stable

substituents[25-27]. Electron-withdrawing groups such as cyanide[22] and fluorine[23,24], and electron-donating groups such as alkoxy[28] decompose at ~300 °C. Additionally, on-surface synthesis is limited to monolayer growth since the dehydrogenation reaction requires contact between prepolymers and a thermally activated metal surface to extract hydrogen atoms[29-31]. For this reason, multilayered growth leading to mass production presents a significant challenge. To overcome these limitations, an original on-surface synthetic approach is required to avoid the thermal decomposition of substituents and to facilitate multilayered growth.

We developed an original technique to produce functional GNRs using electrochemistry (Fig. 1a). Our technique utilizes redox reactions, including ionic polymerization and oxidation-based dehydrogenation, which occur in the electric double layer (EDL) at the liquid-solid interface[32-34]. With EDL, an intense electric field can be confined to a

[1]Institute of Advanced Energy, Kyoto University, Uji 611-0011, Japan. [2]Department of Materials Science and Engineering, Kyoto University, Kyoto 606-8501, Japan. ✉e-mail: sakaguchi@iae.kyoto-u.ac.jp

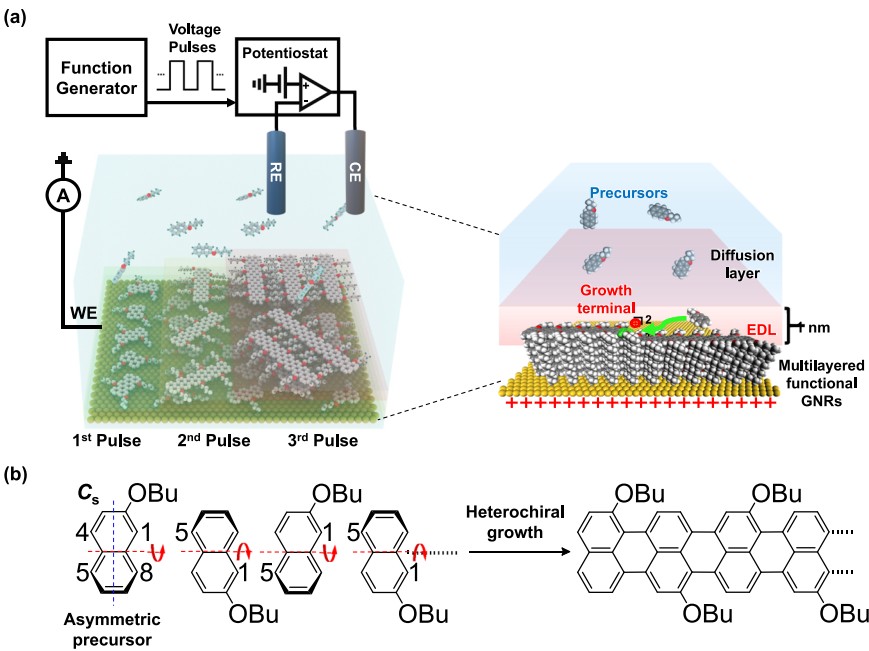

**Fig. 1 | Concept of electrochemical on-surface synthesis.** Illustration of **a** electrochemical on-surface synthesis and **b** heterochiral growth of functional GNRs from asymmetric precursors of 2-butoxynaphthalene.

thickness of approximately 1 nm at the solid-liquid interface. This confinement enables the production of active intermediates that facilitate multilayered growth of GNRs through redox reactions at remarkably low temperatures. Our approach demonstrates heterochiral growth of armchair-edged 5-AGNR with covalently bonded butoxy substituents on an electrode surface, from an electrolyte solution containing the asymmetric ($C_s$ point group) precursor, 2-butoxynaphthalene (Fig. 1b). This is possible due to the low-temperature process of electrochemistry, which enables the production of edge-functionalized GNRs without substituent decomposition. Additionally, the electrochemical approach allows layer-by-layer growth, enabling mass production. This is because the EDL can exist on the top layer even if the electrode surface is fully covered with products[35,36], continuing growth in the vertical direction (Fig. 1a). We discovered this nonlinear mechanism of electrochemical on-surface synthesis, in which the electrochemically generated di-cation by two-electron oxidation at high voltage becomes the active intermediate, responsible for heterochiral di-cationic polymerization followed by oxidation-based dehydrogenation, resulting in formation of GNRs. Electrochemically produced GNRs have a lower oxidation potential compared to tetrathiafulvalene (TTF) indicating strong electron-donating properties. This unique feature makes them highly promising catalysts in silicon etching, and photoconductivity. To date, electropolymerization was the commonly used method to create bulk conducting polymers, by initiating cation-radical polymerization using low voltage[37]. Electrochemical dehydrogenation of polyaromatic hydrocarbon molecules on metal has been previously reported[38]. However, there has been no electrochemical demonstration reported for the synthesis of GNRs. Our developed technique differs from existing electrochemical methods, as it facilitates regioselective ionic polymerization of asymmetric precursors, followed by redox-based dehydrogenation, by applying high voltage with controlled pulse duration. This allows highly organized growth of GNRs on electrodes with molecular-scale precision.

## Results and Discussion
### Structural determination of electron-donating GNRs
A schematic illustration of electrochemical on-surface synthesis is shown in Fig. 1a. To prepare the sample solution, a mixture of 5 mM 2-butoxynaphthalene as the precursor and 0.1 M tetrabutylammonium hexafluorophosphate as the electrolyte was dissolved in anhydrous *o*-dichlorobenzene (*o*-DCB). The sample solution was prepared in a glove box with an environment containing less than 1 ppm of oxygen and humidity. Working electrodes such as Au(111), iodine-covered Au(111)[39,40] and indium tin oxide (ITO)-coated glass were immersed in the sample solution at 80 °C, followed by application of voltage pulses versus the reference electrode using a potentiostat equipped with a function generator (Fig. 1a). Atomic flat Au(111) and iodine-covered Au(111) substrates were utilized for scanning tunneling microscopy (STM) measurements. Iodine-covered Au(111) improves the alignment of GNRs due to lattice matching between GNRs and the iodine atoms on the substrate (Supplementary Figs. 1 and 2). A transparent ITO substrate was used for optical measurements. The reaction temperature of 80 °C enhances the formation of GNRs by overcoming activation energy and improving precursor diffusion, compared to room temperature (Supplementary Fig. 3)[41,42]. The cyclic voltammogram (CV) of the sample solution shows a one-electron oxidation peak at approximately 1.5 V and a two-electron oxidation peak at >2.5 V (Supplementary Fig. 4).

Low-temperature STM (LT-STM) operated at 77 K in a UHV environment was used to identify products electrochemically grown on iodine-covered Au(111). An LT-STM image of electrochemically grown products shows a strand backbone with side groups arranged alternately on both edges (Fig. 2a). Substituents at one edge have a periodicity of 0.85 nm and strands are 1.65 nm wide (Fig. 2b, c). These experimental values agree with those in the proposed model of armchair-edged 5-AGNRs with butoxy substituents. Use of iodine-covered Au(111) results in alignment and elongation of strands of GNRs, in comparison to those grown on Au(111) alone (Supplementary Fig. 1a–d). GNRs grown on iodine-covered Au(111) have an average length of 10 nm, equivalent to a precursor unit number of 11.7. In contrast, those grown on Au(111) are shorter, with an average length of 5 nm. The same Raman spectra are obtained from Au(111) and iodine-covered Au(111), indicating identical products (Supplementary Fig. 1e, f). These findings suggest that iodine-covered Au(111) enhances the orientation of GNRs on the surface (Supplementary Figs. 1 and 2). Lattice parameters of iodine-covered Au(111) in a compressed

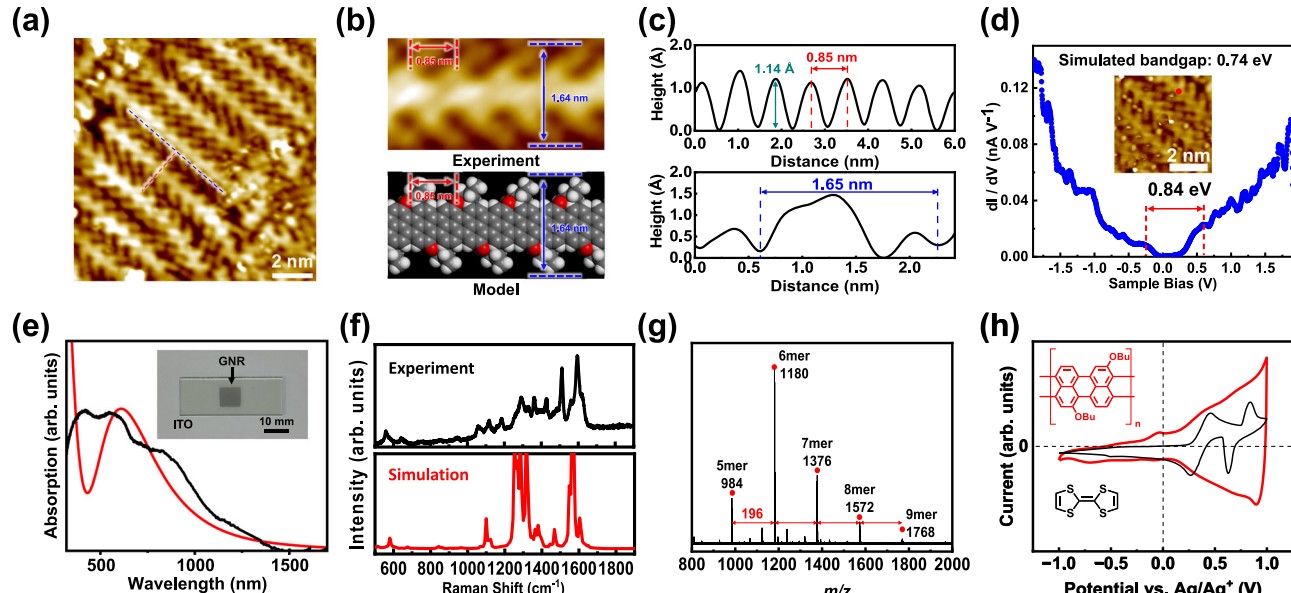

**Fig. 2 | Characterization of electrochemically produced GNRs. a** LT-STM image (−0.97 V, 580 pA) of electrochemically produced sample (5 V, 0.5 s, 3 cycles). Red and blue dotted lines indicate the marks for **b. b** Magnified LT-STM image of the strand with the structure of a GNR. (**c**) Cross-sectional analysis of strand periodicity (top) and width (bottom). **d** STS profile and simulated DFT bandgap. The inset shows the measured location and the simulated bandgap value. **e** Optical absorption spectrum of GNR films on ITO (black) and the simulation (red). The inset displays a photograph of GNR films. **f** Experimental (top) and simulated (bottom) Raman spectra. **g** MALDI-FT-ICR MS spectrum of GNRs showing the number of units. **h** Cyclic voltammetry of GNR films on the Au(111) (red) and TTF (black) in *o*-DCB electrolyte solution. The inset shows chemical structures.

hexagonal structure, along the a, b, and c axis are 4.2, 5.4, and 4.2 Å, respectively, while those of Au(111) are 2.9 Å. The periodicity of GNR (8.5 Å) mismatches with the atom spacing of substrates (Au(111) and iodine-covered Au(111)) by 3.53% and 1.17%, respectively. This indicates that iodine-covered Au(111) is more suitable than Au(111) for epitaxial GNR growth. The large area LT-STM image reveals that GNRs grow uniformly on the substrate (Supplementary Fig. 5). Scanning tunneling spectroscopy (STS) of electrochemically produced GNRs on iodine-covered Au(111) reveals a bandgap of 0.84 eV, while density functional theory (DFT) provides a value of 0.74 eV (Fig. 2d). This minor difference between experimental results and DFT calculations may be attributed to electron screening by a metal substrate and charge transfer between GNRs and the substrate. The bandgap measured by STS is consistent with that measured by optical means, which we will discuss later. Furthermore, we performed d*I*/d*V* mapping of GNR. The experimental d*I*/d*V* mapping agrees roughly with the simulation result. (Supplementary Fig. 6).

The optical absorption spectrum of electrochemically produced GNRs on the ITO electrode exhibits a peak in the near-infrared region, extending to 1500 nm (Fig. 2e). The bandgap of 0.85 eV obtained from the Tauc plot in the absorption spectrum agrees with that obtained from STS (Fig. 2d and Supplementary Fig. 7). Additionally, the Raman spectrum of electrochemically produced GNRs shows G-bands (1600 cm⁻¹), D-bands (1000–1400 cm⁻¹), and radial breathing-like mode bands (544 cm⁻¹) (Fig. 2f). A simulation of Raman was conducted using a Gaussian program on a 10-mer of GNR in a gas-phase. Peaks of G-bands and radial breathing-like mode bands matched the experimental results. However, in comparison to experimental data, the D-band simulation covers the experimental peak positions, whereas the intensity in the simulation deviates from the experimental. This could be due to the dispersion of GNRs with varying lengths. Furthermore, the matrix-assisted laser desorption/ionization Fourier transform ion cyclotron resonance mass spectrum (MALDI-FT-ICR-MS) of electrochemically produced GNRs represents periodic peaks with an interval of 196 m/z, corresponding to the mass of units of the proposed GNR structure (Fig. 2g and Supplementary Fig. 8). Roughness of

the substrate prevents STM measurements of the products on ITO. (Supplementary Fig. 9). Products are confirmed as GNRs due to their Raman spectrum which correspond to those of GNRs produced on Au(111) (Supplementary Fig. 10).

Based on these structural characterization results, we conclude that the chemical structure of the product is an armchair-edged 5-AGNR with butoxy substituents. Our electrochemical technique allows low-temperature edge-functionalization of GNRs while avoiding the decomposition of substituents, unlike high-temperature, on-surface synthesis in UHV[22–24]. The reaction pathway is supposed to involve heterochiral coupling of 2-butoxynaphthalene (Fig. 1b), which is alternatively linked by arranging its aromatic plane up and down, followed by dehydrogenation. We speculate the existence of specific "hot spots" in the active species where coupling reactions occur (Fig. 1b). Furthermore, the CV of electrochemically produced GNRs on Au(111) shows a significantly low oxidation potential at −0.1 V versus Ag/Ag⁺ (Fig. 2h). This value is lower than the first oxidation peak of the strong electro-donating molecules of TTF at 0.4 V, making it one of the strongest electron donors, due to high-density introduction of butoxy substituents per unit into the GNR backbone. It exhibits reversible redox peaks in forward and reverse scans (Fig. 2h), while 5- and 7-AGNR without butoxy substituents produced by two-zone chemical vapor deposition[43] show irreversible features, suggesting decomposition (Supplementary Fig. 11). Electrochemically produced GNRs demonstrate excellent electrochemical cycling durability.

## Mechanism of electrochemical on-surface synthesis

To identify the active species involved in electrochemical on-surface synthesis of GNRs, we conducted electro-absorption spectroscopy of the sample solution using a sandwiched ITO cell with a 250-μm spacer filled with electrolyte solution containing 2-butoxynaphthalene precursors (Fig. 3a and Supplementary Fig. 12). Electro-absorption demonstrates nonlinear behavior with respect to voltage (Supplementary Fig. 13a). No absorption is observed at <2 V. However, strong absorption appears in the infrared region at 1600 nm when the voltage is over 2 V (Fig. 3a and Supplementary Fig. 13b). After turning off the

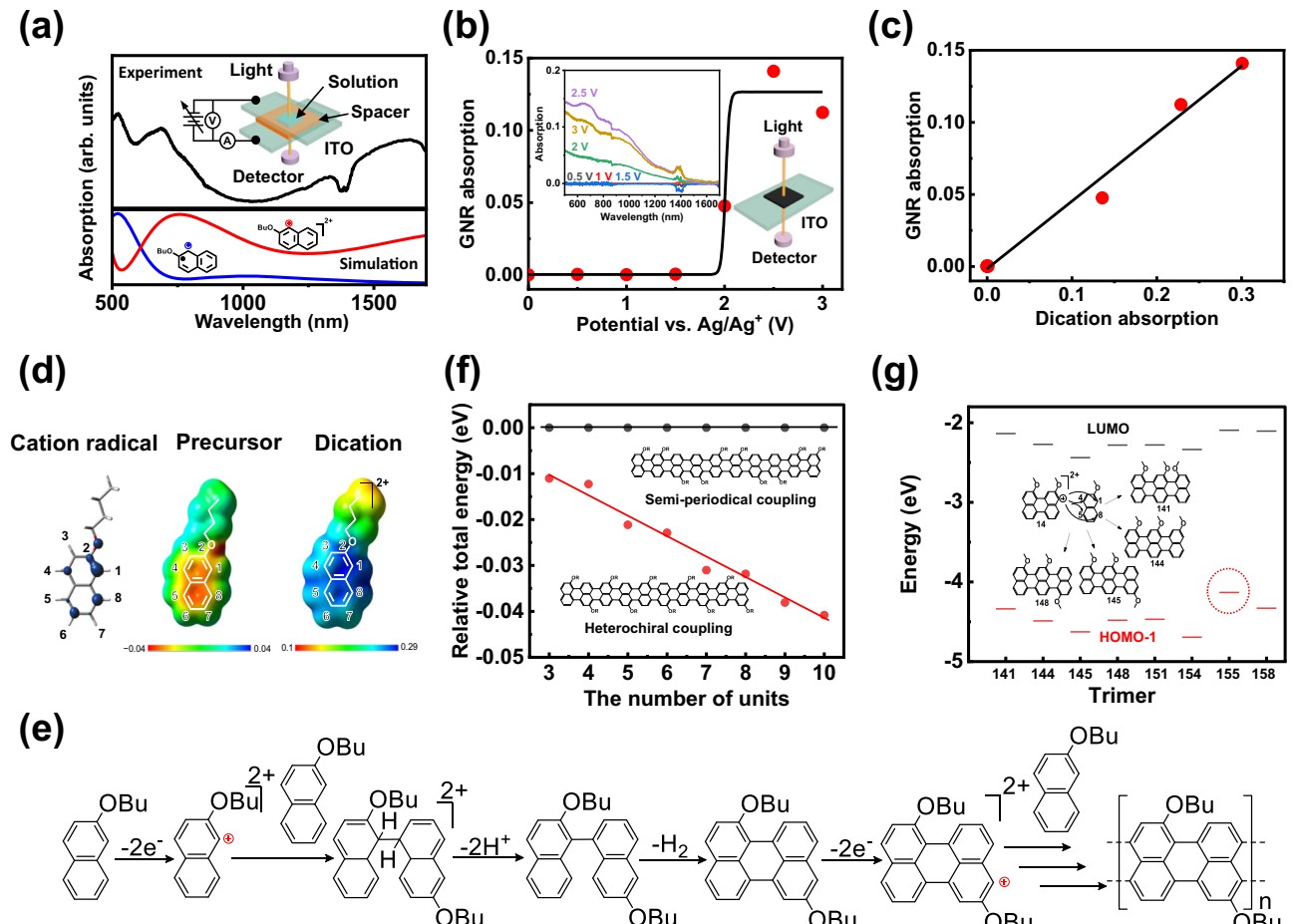

**Fig. 3 | Mechanism of electrochemical growth of GNR. a** Electro-absorption spectrum after voltage application (3 V, 15 s; top) and simulated spectra of di-cation and cation radicals (bottom). The inset shows the experimental setup. **b** The electrochemical voltage dependence on the absorption of a GNR film formed on ITO at 650 nm. The inset shows the setup and voltage-dependent absorption spectra of a GNR film. **c** The correlation between the di-cation absorption at 1600 nm in the cell and GNR absorption at 650 nm in the films. **d** Spin density of the cation radical (left), electron density of precursor (center) and di-cation (right). **e** Reaction mechanism of GNR growth. **f** The unit number dependence of total energies of the heterochiral-coupled GNR (red) and semi-periodical-coupled GNR (black). The inset shows structures. **g** Energy levels of HOMO−1 and LUMO for possible trimer products. The inset shows examples of products. The dotted red circle indicates the highest HOMO−1.

voltage, infrared absorption decays within 50 s, indicating active intermediates (Supplementary Fig. 14). DFT calculations predict that the di-cation resulting from two-electron oxidation of the precursor exhibits infrared absorption above 1500 nm, whereas the cation radical of the precursor resulting from a one-electron oxidation shows near-infrared absorption around 1000 nm (Fig. 3a). The electro-absorption spectrum of the solution applied at a voltage of 3 V agrees with the simulation of the di-cation of precursors (Fig. 3a). Additionally, the voltage dependence on the growth yield of GNR was studied by analyzing the optical absorption of the GNR film formed on the ITO electrode (Fig. 3b). The results indicate a nonlinear response similar to that of the solution (Supplementary Fig. 13b). Consequently, the yield of GNR is proportional to the amount of electrochemically generated di-cation species (Fig. 3c). These results suggest that the di-cation is the active species that responds to GNR growth.

To gain insight into the electrochemical growth mechanism of GNR, we conducted theoretical studies on the thermodynamics and kinetics. Initially, we identified the chemical species that contribute to growth reactions among precursors, radical cations, and di-cations. Two growth mechanisms can be considered, the radical cation mechanism and the di-cation mechanism. The electropolymerization of conducting polymers, such as polythiophenes, adopts the radical cation mechanism, where coupling occurs among radical cations at

positions with high spin density[44]. This mechanism is invalid for heterochiral coupling due to the absence of the required high spin density of the radical cation on carbons at positions 1 and 5 (Fig. 3d). In the di-cation mechanism, the di-cations generated by two-electron oxidation of the precursor might serve as electrophilic agents, driving the coupling reaction by attacking precursor molecules (Fig. 3e). The DFT-calculated electrostatic map indicates that the carbon at position 1 in the di-cation has the highest hole density, suggesting that it is an active site, whereas other positions have delocalized densities (Figs. 3d, 1b). This is in sharp contrast to the precursor molecule, in which all carbons in 2-butoxynaphthalene have a featureless electron density mapping (Fig. 3d).

Next, we conducted a thermodynamic study using DFT calculations to find the pathway of GNR growth based on the di-cation mechanism. We explored potential combinations of reactive sites in coupling reactions between the di-cation and precursor and compared the free energy of the resulting coupled products. We generated products for dimers by coupling the carbon at position 1 of the di-cation with carbons at positions 1, 4, 5, and 8 of the precursor to explore all possible chemical structures of dimers (Supplementary Fig. 15). The same method was also used to generate trimer and tetramer products (Supplementary Figs. 16 and 17). In our model, we assume that dehydrogenation reactions coincide with polymerization

reactions during GNR growth. Due to their lower oxidation potential, oligomers and polymers may undergo one-electron oxidation more easily than precursors, creating C-C bonds through dehydrogenation. Based on DFT calculations, the minimum free energy for the dimer, trimer, and tetramer is achieved by heterochirally coupled GNR products formed between the carbon located at position 1 of di-cation and carbon located at position 5 of precursor (Supplementary Figs. 16 and 17). The length dependence on the free energy of heterochirally coupled GNRs was compared to that of GNR coupled at other positions, revealing a significant negative slope (Fig. 3f). These results show that the heterochirally coupled GNR is thermodynamically stable and experienced less strain compared to other GNRs. From a kinetic standpoint, we compared energy levels of HOMO-1 for possible coupled GNR products at various positions for dimers and trimers using DFT calculations, representing the trend of two-electron oxidation to form di-cations. The study shows that GNR products with heterochiral coupling have the highest HOMO-1 energy levels compared to other GNR products (Fig. 3g and Supplementary Fig. 18). These results suggest that through heterochiral coupling GNR products undergo efficient two-electron oxidation, converting them into the di-cation form and promoting di-cationic polymerization reactions.

Our proposed di-cation model explains the selective reaction between carbon at position 1 of di-cation and carbon at position 5 of the precursor, leading to heterochirally linked edge-functionalized GNRs (Fig. 3e). Our model is also supported by evidence of the precursor effect on electrochemical GNR growth using symmetric precursors like 2,6-dibutoxy and 2,7-dibutoxynaphthalenes (Supplementary Figs. 19 and 20). According to STM measurements (Supplementary Fig. 19), these symmetric precursors are unable to produce GNRs under the same electrochemical conditions using 2-butoxynaphthalene. Moreover, from electro-absorption spectroscopy measurements, they exhibit no electrochemically generated di-cation species (Supplementary Fig. 20). These results suggest that the symmetry of precursors is crucial in electrochemical on-surface synthesis of GNRs. Asymmetric precursor structure in 2-butoxynaphthalene can stabilize the hole created in the carbon at position 1 of di-cation, while symmetric precursors cannot. As conclusion, the electrochemical growth of GNRs occurs through heterochiral di-cationic polymerization, where the carbon at position 1 of the GNR terminal undergoes nucleophilic attack and couples with the carbon at position 5 of the precursor (Fig. 3e). This reaction occurs for both thermodynamic and kinetic reasons. The GNR growth mode is believed to be chain-growth because the di-cation active site at the growth terminal is not able to diffuse on the surface but binds tightly to it.

## Electrochemical layer-by-layer growth of GNRs

LT-STM was used to investigate the GNR growth process on iodine-covered Au(111) by increasing the number of applied voltage pulses. When a single voltage pulse of 5 V is applied for 0.5 s to the electrode, the resulting LT-STM image displays a random arrangement of short chains, all less than 1 nm in length (Fig. 4a). The coverage ratio of this sample is estimated to be about 82.3%. After a second pulse, a few GNR strands with longer chains (around 10 nm) appear in the LT-STM image (Fig. 4b). A third pulse results in a fully covered electrode with a highly ordered array of GNRs with long chains of 10 nm (Fig. 4c). The fourth pulse produces a high-density array of GNRs similar to the third pulse, but with a little bit of disorder (Fig. 4d). LT-STM images (Fig. 4c, d) show that the majority of the strands are composed of linear chains that are coupled in a regioselective manner. However, kinks are sometimes observed at the ends of these straight chains, suggesting that there may have been collisions with neighbouring chains during their growth. This collision indicates that coupling is non-regioselective.

In order to investigate the thickness of multilayered GNRs, AFM measurement of transferred GNRs from iodine-covered Au(111) to Si substrate was carried out because STM cannot characterize the thickness of GNR films due to substrates fully covered by GNRs (Supplementary Figs. 21 and 22). It was observed that the thickness and Raman intensity of GNR films were directly proportional to the number of voltage pulses applied. As per the STM characterization, the estimated thickness of a monolayer was 0.114 nm (Fig. 2c). When a voltage pulse of 5 V was applied, a 2.46-layer was formed as per the AFM characterization (Fig. 4e and Supplementary Fig. 22). These results imply that the electrochemical on-surface method allows growth of multiple layers, unlike most types of UHV on-surface synthesis, which is limited to monolayer growth based on metal catalytic dehydrogenation that requires contact between reactants and metal surfaces[29-31]. One exception, involving bilayered GNRs was reported using UHV on-surface synthesis[45]. This method accomplishes the dehydrogenation reaction of prepolymers, originating from a domino-like effect remotely triggered by the direct contact between the GNR segment and the gold substrate. Direct contact between the prepolymer and a metal substrate is necessary for second-layer growth, which still remains a limitation of multilayer growth. In contrast, electrochemical on-surface synthesis is based on redox reactions, such as polymerization and dehydrogenation, which occur at the solid-liquid interface, EDL[32-34]. Electrochemical growth in the EDL allows layer-by-layer deposition of GNRs on the electrode surface. This is because the EDL is always present on the top layer[35,36], even if the electrode surface is fully covered by GNRs (Fig. 1a). In LT-STM snapshots (Fig. 4), it is clear that elongated GNR growth begins on the electrode covered by GNRs with short chains. These observations suggest that GNRs adhere more effectively to adlayers than to the electrode surface. It's possible that adlayer structural matching with GNRs improved the ordered arrangement of strands in 3rd pulses. It has been reported that the ordering of poly-*p*-phenylene (PPP) is driven by surface-bound bromine[46]. Additionally, other reports suggest that polythiophene's orientation is enhanced by iodine-covered Au(111)[40]. These phenomena are attributed to epitaxy, where polymers grow along the crystalline lattice on the substrate due to interactions between halogen atoms and polymers. In this study, the alignment of GNRs is expected to be improved by epitaxy based on π-π interactions between GNRs on the top layer and the short chain of GNRs in the underlayer. This unique aspect of electrochemical on-surface synthesis makes it an invaluable technique for mass-producing edge-functionalized GNRs.

## Functionalities of electron-donating GNRs

We explored applications of electrochemically produced GNRs, which exhibit strong electron-donating properties. Metal-assisted chemical etching (MACE) is a process in which metal is used to help create complex structures in silicon by scissoring bulk materials[47-49]. This technique has great potential in the semiconductor industry as it provides a method of creating Si nanoarchitecture[47-49]. Noble metals like gold, silver, and platinum are excellent catalysts for MACE with high etching rates. However, they have limitations, including limited availability and the possibility of device contamination. To overcome these drawbacks, carbon-based catalysts like carbon nanotubes (CNT) and graphene have gained much attention[50-52]. However, these materials have inefficient etching rates compared to noble metals. We investigated the catalytic activity of electrochemically produced GNRs for Si-etching. Si etching experiments were conducted by transferring GNRs grown on electrodes to Si substrates (Supplementary Fig. 21) and exposing them to a vapor mixture of hydrofluoric acid and hydrogen peroxide at 50 °C (Supplementary Fig. 23)[53]. The region with deposited GNRs undergoes significant Si etching, while the area without GNRs remains unetched (Fig. 5a–c and Supplementary Fig. 24). The smallest etched area was 1 × 2 μm (Supplementary Fig. 25). This reveals that

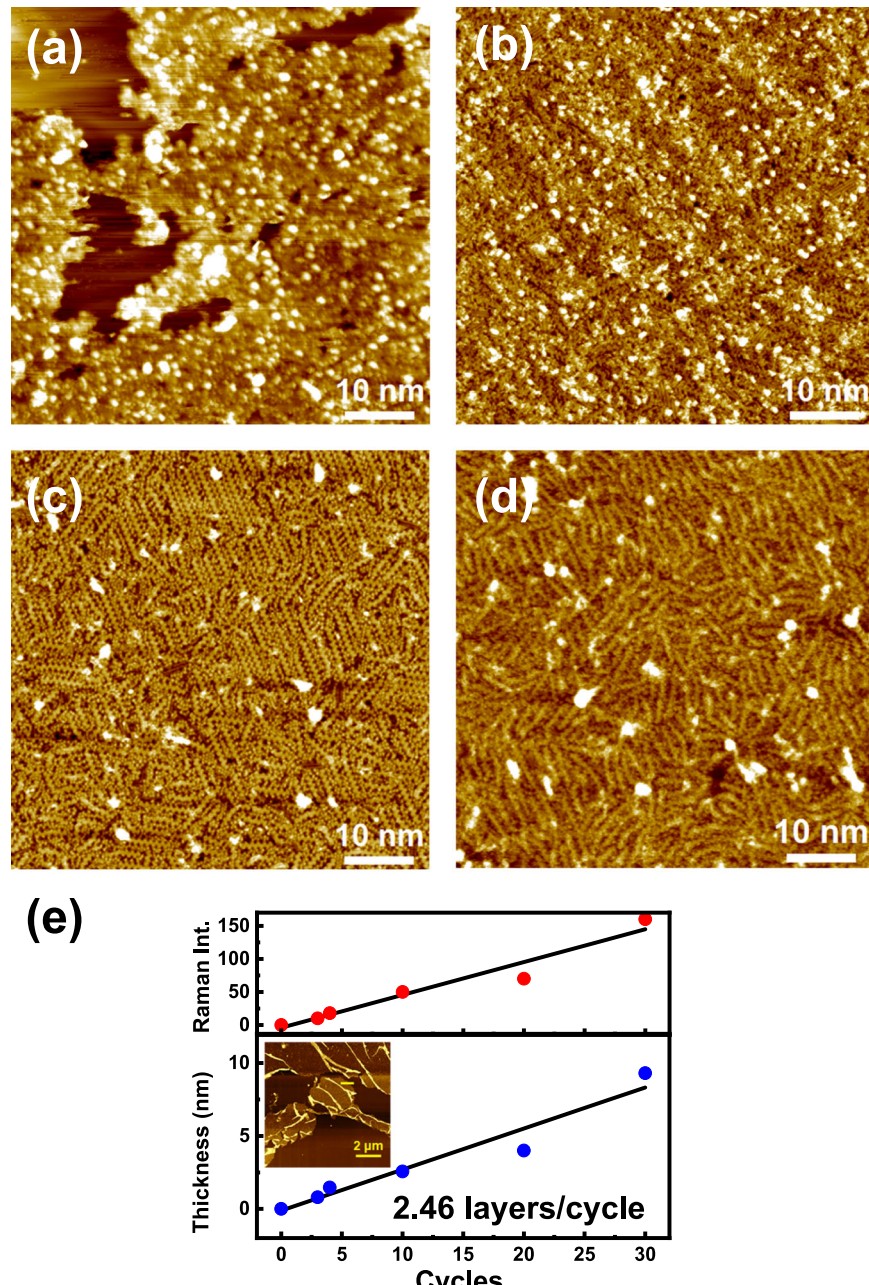

**Fig. 4 | Voltage cycle dependence on electrochemical GNR growth.** LT-STM images of electrochemically produced samples (5 V, 0.5 s) with **a** one cycle (STM measurement conditions: −1.95 V, 160 pA), **b** two cycles (−0.31 V, 580 pA), **c** three cycles (−0.76 V, 580 pA) and **d** four cycles (−0.81 V, 580 pA), respectively. **e** Thickness and Raman intensity of electrochemically produced GNR films transferred from iodine-covered Au(111) to Si substrates with different numbers of cycles (5 V, 0.5 s). The inset shows an AFM image of a GNR film (30 cycles) on Si.

electrochemically produced GNRs exhibit a high etching rate of 2.0 μm h⁻¹, which is superior to that of gold (1.4 μm h⁻¹) and reported to be the most effective catalyst to date (Fig. 5d). We also confirmed that precursor did not etch Si substrate. (Supplementary Fig. 26). Moreover, compared to other carbon-based catalysts such as metallic-CNT (0.1 μm h⁻¹), semiconductor-CNT (0.4 μm h⁻¹), graphene (0.1 μm h⁻¹), 5-AGNR (0.8 μm h⁻¹) and 7-AGNR (0.2 μm h⁻¹), electrochemically produced GNRs exhibit the highest etching rate, making them an excellent Si-etching catalyst (Fig. 5d and Supplementary Fig. 27). The CV of electrochemically produced GNRs displays a low oxidation potential of −0.1 V vs. Ag/Ag⁺, and a broad oxidation spectrum (Fig. 2h). High electron-donating semiconductor properties of electrochemically produced GNRs can enhance Si-etching performance. This may be due

to the efficient reduction of hydrogen peroxide at a potential of 1.78 V vs. SHE, followed by hole transport to the shallow level in the valence band, and finally Si oxidation at 0.67 V vs. SHE (Supplementary Fig. 28).

Moreover, we investigated photoconductive properties of electrochemically produced GNRs. To prepare the sample cell, we doped an electron acceptor, FeCl₃, into an electrochemically produced GNR film on a glass substrate, and then deposited gold with a 25-μm gap (Supplementary Fig. 29). When the sample cell was irradiated with a laser at 532 nm with a power of 1 mW, we observe intense photocurrent peaks with an ON/OFF ratio of 1.68 (Fig. 5e). As part of the control experiments, the photoconductive behavior of 5-AGNR and 7-AGNR, which were produced by chemical vapor deposition[43], was measured. However, no detectable photoconductive signals were observed in

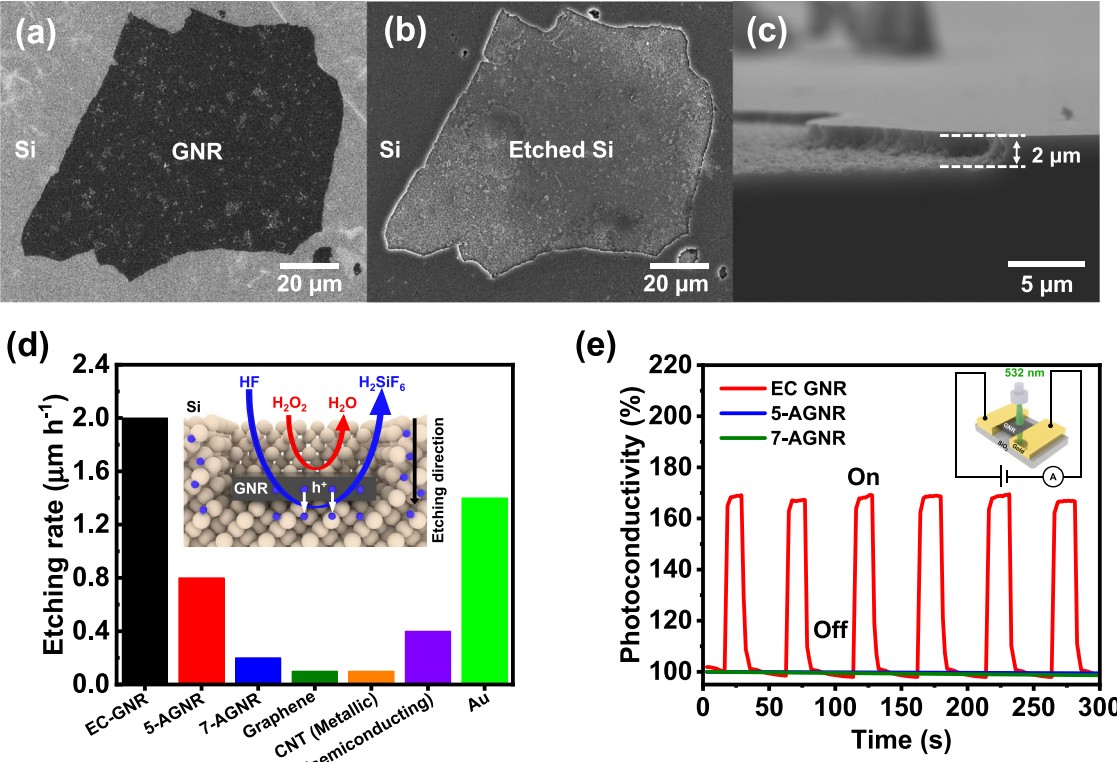

**Fig. 5 | GNR-assisted chemical etching of Si and photoconductivity.** SEM images of GNR-covered Si **a** before and **b** after vapor-phase chemical etching by HF and $H_2O_2$. **c** Cross-sectional image of **b**. **d** Catalytic performance of Si etching of electrochemically produced GNR, 5-AGNR, 7-AGNR, graphene, metallic CNT, semiconducting CNT and gold. The inset shows the mechanism of GNR-assisted Si etching. **e** Photoconductivity of electrochemically produced GNR, 5-AGNR, and 7-AGNR with $FeCl_3$ doping. The inset shows the experimental setup.

cells of such GNRs (Fig. 5e). Additionally, we confirmed that electrochemically produced GNRs without $FeCl_3$ doping did not show photoconductivity (Supplementary Fig. 30). These results indicate efficient generation of photo-carriers in GNRs electrochemically produced with electron transfer from photoexcited states of GNRs to $FeCl_3$, which acts as an electron-acceptor, resulting in hole transport. The introduction of butoxy moieties into the GNR backbone of 5-AGNRs provided a significant electronic advantage over hydrocarbon-based GNRs.

On-surface synthesis of GNRs is based on chemical reactions occurring on a thermally activated metal surface in a UHV environment. This process leads to the creation of various types of GNRs. However, gas phase reactions on metal surfaces that are driven by neutral species via radical polymerization and metal catalytic dehydrogenation require high temperatures. Our electrochemical on-surface technique has unique features such as low-temperature growth and layer-by-layer multilayer growth on the electrode. These features are believed to result from redox reactions, including heterochiral di-cationic polymerization and oxidation-based dehydrogenation reactions in EDL. As a result, the synthetic temperature of GNRs is remarkably reduced. Our technique raises the possibility of producing various edge-functional GNRs, even with thermally unstable substituents.

From the perspective of the polymerization mechanism, conventional active species such as radicals, radical cations, and cations have been involved in reported polymerization reactions[54–56]. Our discovery of heterochiral di-cationic polymerization is unique because it involves the use of di-cation forms of asymmetric precursors. Applying high voltage results in the production of a growing polymer terminal with high hole density at specific positions, leading to regioselective polymerization in an ionic mode. Nonlinear

mechanisms in polymerization can lead to unique polymer structures.

Our electrochemical on-surface synthesis can be applied to various aromatic precursors, leading to wider GNRs. Moreover, electrochemically produced GNRs present excellent silicon-etching properties compared to noble metals, implying great potential for Si nano-lithography and exhibiting excellent photoconductivity.

## Methods

### Electrochemical on-surface synthesis
A sample solution was prepared in a glovebox by mixing 5 mM of 2-butoxynaphthalene and 100 mM of tetrabutylammonium hexafluorophosphate in o-DCB. Working electrodes, including Au(111), iodine-covered Au(111) and ITO, 0.5 × 1.0 cm² in size, were used. Platinum and silver wires were used as counter and reference electrodes, respectively. The sample cell was heated at 80 °C on a hot plate with stirring. Voltage pulses were applied using a potentiostat (Huso Electro Chemical System, HECS 990 C) equipped with a function generator (NF, Wave Factory WF1974).

### Electro-absorption measurement
In the electro-absorption cell, ITO electrodes were sandwiched with a plastic spacer of 250 μm (Supplementary Fig. 12). To perform electro-absorption spectroscopy, we injected 50 μL of sample solution into the cell. We used an optical absorption spectrometer (Lambda Vision Inc., LVmicro/SUV-100S-NIR) for this purpose (Fig. 3a, c, Supplementary Figs. 13 and 14). To measure potential-dependent electro-absorption, we applied various voltages ranging from 0.5 V to 3 V for 15 s (Supplementary Fig. 13). For the temporal profile of electro-absorption measurements, we applied 2.5 V for 25 s (Supplementary Fig. 14) and recorded from start to 55 s.

## Mass spectroscopy

The sample was analyzed using high-resolution mass spectrometry (HRMS) in positive ion mode. The instrument used was a Fourier transformation-ion cyclotron resonance mass spectrometer (FT-ICR-MS), equipped with a 7-tesla superconductive magnet and a matrix-assisted laser desorption/ionization (MALDI) ion source. The matrix used was *trans*−2-[3-(4-*tert*-butylphenyl)−2-methyl-2-propenylidene] malononitrile (DTCB).

## Scanning tunneling microscopy (STM) and scanning tunneling spectroscopy (STS)

All samples were degassed in UHV with annealing at 150 °C for 1 h before STM measurements. STM measurements were conducted in constant-current mode using a low-temperature scanning tunneling microscopy system (UNISOKU, USM-1100SA-2C) operating at 77 K. The tip was a tungsten wire that had been electrochemically etched. d$I$/d$V$ spectra were acquired under a reduced feedback loop during voltage sweeps with $V_{sample}$ set at −0.5 V and $I$ at 200 pA. The differential conductance map was recorded with set $V_{sample}$ set at −0.2 V or 0.6 V and/at 80 pA by a digital lock-in amplifier with a modulation frequency of 350 Hz and an AC voltage of 2 mV. Images obtained were analyzed using SPIP software.

Supplementary Figs. 1 and 3 were obtained using an instrument (formerly Molecular Imaging, PicoSPM) under Ar at room temperature. STM measurements were performed in constant-current mode, with all STM images taken at a tip bias of 0.2 V and a constant current of 5 pA. The tip used was an electrochemically etched Pt-Ir (80:20) wire.

## Material-assisted chemical etching of Si

To test electrochemically produced GNRs, we applied voltage pulses (3 V, 5 s, 2 cycles). After the process, we washed the resulting GNR-covered gold substrate with DCM before transferring it. We then etched the gold substrate using a solution of 50 mM $I_2$ and 1.1 M KI in water. To transfer the GNR, we placed the Si substrate (0.5 × 1.0 cm$^2$) on top of the etching solution, where it covered the GNR. The substrate was washed with $Na_2SO_3$ solution and then with deionized water to quench $I_2$ (Supplementary Fig. 21).

To conduct etching experiments (Fig. 5a−c and Supplementary Fig. 23), the following procedure was employed. Initially, a solution of 25 mL of 46% HF and 0.5 mL of 30% $H_2O_2$ was prepared and injected into a sealed Teflon container. The container was then preheated in a water bath for 30 min at 60 °C. Next, the Si covered with GNR was placed on the platform inside the container, ensuring no direct contact with the etchant solution. The container was sealed and heated in the water bath in a dark environment at 50 °C for 1 h. Finally, the Si was washed with deionized water.

To test CNT and graphene, we dropped 10 µL of sample solutions on Si substrates and then dried them at 60 °C for 30 min. We used commercially available solutions of metallic CNT dispersed (Meijo nano carbon, EC-DL) and semiconducting CNT dispersed (Meijo nano carbon, RS-1), which were diluted 10x with deionized water, to create CNT solutions. For the graphene solution, we sonicated graphene (FUJIFILM Wako Chemicals) that was dispersed in 1-methyl-2-pyrrolidone (0.1 mg mL$^{-1}$) for 30 min and then centrifuged it. We used the supernatant to create the solution. To create Si substrates covered by 5-AGNR and 7-AGNR, we followed the same transfer procedure as for electrochemically produced GNRs (Supplementary Fig. 21).

## Atomic Force Microscope (AFM) Characterization

We prepared samples to determine the thickness of GNRs using electrochemical means on an iodine-covered Au(111) by applying voltage pulses of 3, 4, 10, 20, and 30 at 5 V for 0.5 s. The GNRs were then transferred to the Si substrate using the transfer method. To investigate the thickness of GNRs on the Si substrate, we performed AFM measurements (CSI, AFM Galaxy Dual Controller, and Digital Instruments, MultiMode AFM-2) (Supplementary Fig. 22).

## Photoconductivity

To measure photoconductive properties, we prepared GNRs electrochemically on an ITO substrate by application of voltage pulses (3 V, 0.5 s, 3000 cycles). The resulting GNRs on ITO were then doped by immersing them in a 10 mM $FeCl_3$ acetonitrile solution for 2 min in a glovebox, followed by several washes with acetonitrile and DCM (Supplementary Fig. 29). After doping, $FeCl_3$-doped GNRs on ITO were peeled off the substrate by sonicating it in DCM. Collected flakes of GNRs in DCM solution were then transferred onto glass substrates and dried on a hot plate at 80 °C for 10 min. To create a photoconductive cell, gold was deposited on transferred GNRs on a glass substrate, which was covered with a 25-µm grid mesh. Photoconductive cells of 5-AGNR and 7-AGNR were prepared the same way as electrochemically produced GNRs.

The semiconductor analyzer (Keithley 4200-SCS) and a home-made probe station were used to measure photoconductivity properties. Once the sample was connected to the probes (Fig. 5e), the chamber was evacuated to $10^{-4}$ Pa. Current was measured at a constant voltage of 1 V in a dark environment. The sample was exposed to 6 cycles of a green laser at 532 nm with a power of 1 mW for 20 s, with a 30 s interval between cycles.

## Simulation of absorption spectra

The purpose was to identify species that participate in the polymerization reaction, such as cation radicals and di-cations of 2-butoxynaphthalene. TD-DFT calculations were conducted at the TD-UB3LYP[57,58] level using a 6-311 G(d,p) basis set. A Gaussian program[59] was used to simulate absorption spectra of these species.

## Raman simulation

Density functional theory in a Gaussian program[59] was used to simulate the Raman spectrum of a 10-mer under the conditions of B3LYP and the 6-311 G(d,p) basis set.

## Spin density, electron density mapping calculations

To identify chemical species that contribute to growth reactions among precursors, cation radicals, or di-cations of 2-butoxynaphthalene, we carried out calculations for spin density, and electron density mapping calculations using the Gaussian program[59]. The 6-311 G(d,p) basis set was used to calculate the spin density of the cation radical species at the UB3LYP[44] level.

## Total energy and energy level calculations

In order to understand the reaction mechanism in terms of thermodynamics and kinetics, we employed the Gaussian program[59] to compute the total energy and HOMO-1 energy levels of various structures at RB3LYP level using the 6-311G(d,p) basis set. This was done after geometrical optimization. All possible structures of dimer, trimer, and tetramers are shown in Supplementary Figs. 15−17. For example, when position 1 of the di-cation is coupled with position 1 of precursor (neutral species) followed by dehydrogenation reactions, compound **11** is formed (Supplementary Fig. 15). In a similar manner, dimers **11**−**18**, trimers **141**−**158** (Supplementary Fig. 16), and tetramers **1441**−**1558** (Supplementary Fig. 17) are generated.

# Data availability

The data presented in this study are available on request from the corresponding author. The data generated in this study have been deposited in the Zenodo database [https://zenodo.org/records/12566342].

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

## Acknowledgements

This study was supported by KAKENHI Program No. 22H01891 (H.S.), 22K18944 (H.S.), 23K04521 (T.K.), 21K05038 (S.N.), Japan Society for the Promotion of Science, Japan; Zero-Emission Energy Research (ZE2022B-07), IAE, Kyoto University (KF). Computational resources were provided by the supercomputer system at the Institute for Chemical Research, Kyoto University. We thank Atsuya Tada and Zexiao Li for their technical assistance.

## Author contributions

H.S. supervised all experiments and wrote the manuscript. T.K., Y.C. and S.N. performed synthesis, characterization, and theoretical studies. YC and KF performed Si etching and SEM measurements. All authors participated in scientific discussions.

## Competing interests

The authors declare no competing interests.
