## [Peer Review File · Nature Communications]

Electrochemical on-surface synthesis of a strong electron-donating graphene nanoribbon catalystReviewers' Comments:

Reviewer #1:

Remarks to the Author:

In this work, the authors present an electrochemical synthesis/growth of graphene nanoribbons (GNRs) on substrates using the naphthalene compound substituted with an electron-rich butoxy group. The redox behavior of this precursor compound is demonstrated to be responsible for the so-called heterochiral polymerization involving dicationic mechanism followed by the dehydrogenation under the electrochemical condition with voltage. As a result, the authors show the formation of N=5 GNR with edge substitution on the surface with layer-by-layer deposition. Moreover, the authors demonstrate the functionality of the resultant GNRs. Overall, the proposed electrochemical synthesis method for the GNR involving novel precursors and mechanisms is quite interesting and can broaden the current synthetic scopes for the GNR chemistry. Nevertheless, the following comments shall be carefully addressed before further consideration of the current work:

Major comments:

-what is the exact role of the substrate during the electrochemical synthesis in this work? The authors used the Au (111), I₂ passivated Au, and ITO substrates, respectively. The authors also claimed the crucial role of the electrical double layer (EDL) at the liquid-solid interface, which has a thickness of about 1 nm. This also raises the question of how thick (how many layers) of GNRs can be grown with the EDL confinement with this technique?

-what is the influence of temperature on the GNR growth in this work? The authors performed the experiments at 80°C. Does the temperature mainly influence the polymerization step, or the dehydrogenation step, or both? The GNRs grown on the substrates are generally too short (< 10 nm, and it is not clear why I₂-covered Au substrate gives a longer ribbon); any specific reason for this? This would refer to the polymerization step. Then what is the critical role of achieving a high polymerization degree with the precursor monomer and electrochemical method?

-It would be valuable if the authors could briefly discuss the polymerization mechanism with the proposed dication polymerization. Is this the chain-growth polymerization or step-growth polymerization? The different mechanisms also influence the homogeneity of the GNRs.

-in page 4, the authors obtained an optical bandgap of 1.0 eV for the grown GNRs on the ITO substrate and claimed that this value agrees with the result from STS measurement (0.84 eV on the iodine-covered Au (111)). Nevertheless, these two values are still quite deviated from each other. Why? Can the latter be due to the coupling between the GNRs and the Au substrate?

-For the ITO substrate-grown GNRs, the exact structural proof is somehow lacking (only UV and Raman). Did the authors compare these results obtained for both ITO-substrate and Au-substrate grown samples? From Figure 2d, it is not clear to me why there is a strong band at around 1400 nm absorption.

-Figures 4 and S3 show that many GNRs appear kinked (rather than straight); why? Could it be due to the missed/non-selective coupling reaction of precursor monomers during the polymerization step?

-For the photoconductive property study of GNRs on the substrate, why was FeCl₃ doping employed? How about the device performance without FeCl₃ doping?

Minor comments:

-in page 6, the authors claimed the "GNRs demonstrate exceptional chemical stability". It is not clear to me how the chemical stability was evaluated in this work.

-it would be better to call the "electron-donor" to "electron-rich" in the context for the GNRs?

-do I understand it correctly that the dication is fully distributed (rather than being located on certain carbon atoms) on the naphthalene compound in Figure 3d? Of course, this makes it difficult to judge which position of carbon atom in naphthalene would be more favorable for the reaction. The authors shall also label the positions "1 and 5" in the Figure 3d.

-The quality and readability of Figure S13 shall be improved.

-the two important review papers on the GNRs shall be cited (Chem. Soc. Rev. 2015, 44, 6616; Acc. Chem. Res. 2022, 55, 23, 3322).

Reviewer #2:

Remarks to the Author:

In the manuscript "Electrochemical on-surface synthesis of a strong electron-donating graphene nanoribbon catalyst" Hiroshi Sakaguchi and co-workers present an interesting electrochemical approach for the synthesis of graphene nanoribbons equipped with side groups. The authors show the synthesis of well-shaped GNRs containing side butoxy substituents. I believe that the achievement is interesting and appealing for a broad readership. Indeed the on-surface approach suffers from the inability of multilayer generation and lack of resistance of a range of substituents to high temperature treatment needed for reaction initiation. Presented approach provides an interesting step forward. I recommend the manuscript for publication in Nature Communications after minor revision addressing the following points:

- the authors state (e.g. in abstract) that they successfully demonstrate highly organized GNR, I am not convinced that one could describe the system as "highly organized"
- the manuscript needs some language corrections, whereas in general it is clearly written some sentences are difficult to follow, below I attach a few examples - in abstract "Electron-donating GNRs exhibiting one of the strongest electron-donating properties known, enable extraordinary performance on silicon-etching catalytic activities better than noble metals, with superior photoconductive properties.", page 2 : "The redox potential of electrochemically produced GNRs shows one of the strongest electron-donating substances discovered.", page 9: "This result suggests that the oligomers of GNRs may cover the electrode." The above mentioned sentences are unclear. I suggest careful reading and rephrasing especially too long sentences.
- what is the coverage (estimation) in Fig.4a?
- were the samples degassed after transferring to UHV?
- refs 50-51 appear under data availability
- in page 4 the authors say "Additionally, the Raman spectrum of electrochemically produced GNRs shows the G-bands (1600 cm⁻¹), D-bands (1000-1400 cm⁻¹), and radial breathing-like mode bands (544 cm⁻¹) (Fig. 2e). These peaks match those obtained from DFT simulation based on our GNR model", whereas in my opinion the experimental data shown in Figure 2e differs quite substantially from the calculated one. I would suggest to discuss the issue in more detail.
- the data shown in Figure 4 are acquired with very different STM settings (mainly voltage), why?
- can the authors estimate the number of layers in Fig. 4d?
- the etching approach seems very interesting, do the authors know the limit of patches size that could be still applied for etching (the smallest patches)?
- would the precursor molecules also provide etching properties (without formation of GNRs)?
- the patches used for etching seem to be irregular, is there any idea to prepare them in a regular manner with desired dimensions?
- the authors discuss the mechanism of electrochemical synthesis of GNRs, could the approach be easily transferred for the synthesis of e.g. wider GNRs?

Reviewer #3:

Remarks to the Author:

In the manuscript, Sakaguchi et al. reported a novel electrochemical on-surface synthesis approach and obtained multilayered 5-AGNRs which are doped by electron-donating groups. The conventional on-surface synthesis of GNRs often needs a high-temperature annealing and could typically get only monolayered GNR products. The electrochemical method reported here thus provides an exciting pathway for the large-scale fabrication of high-quality GNRs, which could be used for the real applications in nanodevices, as also demonstrated by the authors. The data is of high quality and the majority of conclusion are well supported by a series of control experiments and DFT calculations. Therefore, I would recommend its publication in Nat. Commun. once a few technical points and

drawbacks are satisfactorily addressed.

1. The paper is a little bit too long. I suggest the authors shorten some less important sections to make the manuscript more concise, which will also make the readers easier to grasp the main meaning of the paper.
2. In the introduction section and also some positions of the main text, the authors stated that on-surface synthesis is limited to monolayer growth. This is actually not always true, e.g. *J. Am. Chem. Soc.* 2023, 145, 10126-10135.
3. There are more examples about edge-functionalization of GNRs via on-surface synthesis, such as ketone and amino functionalization, e.g. *Nat. Chem.* 2022, 14, 1451-1458; *Nano Lett.* 2022, 22, 164-171; *ACS Nano* 2020, 14, 1895-1901. The ketone groups could even possibly bring in magnetism to a GNR system. The author should consider to cite these works.
4. The dI/dV STS shown in Fig. 2c is not convincing. The authors should measure a spectrum on a bare Au(111) surface using the same tip to compare. This would help one to know that the resonances in STS are really contributed by the GNR or simply from the tip itself. If a STS on a bare Au(111) is not possible to measure because the surface is fully covered by GNR, the authors should try to map out the resonances in dI/dV spectrum, i.e. dI/dV maps at ~ -0.2 and $+0.6$ V to make sure they are "real" conductance resonances of GNRs.
5. How did the authors perform DFT calculations for the simulation of a bandgap? In gas phase or on a surface? The result from the former cannot be directly compared to the experimental values because the electron screening effect, charge transfer, etc, are not considered. The difference could be very large.
6. Fig. 2b is too small. A good match between the molecular model and the corresponding STM image is the most important evidence to confirm the formation of the 5-AGNR functionalized by OBU groups. Please make it larger.
7. Fig. 3: Please mark the position of site 1 and 5 near the molecular structure. Readers with a poor chemistry background cannot identify the numbers of carbon sites easily.
8. Page 8. "According to the STM measurements (Supplementary Fig. 15), these symmetric precursors are unable to produce GNRs under the same electrochemical conditions using 2-butoxynaphthalene." These STM images were obtained at ambient pressure. Therefore, it is reasonable to argue that the absence of GNR is simply due to the low resolution or a low coverage.
9. As for the iodine-covered Au(111), the authors demonstrated that the GNR in the second and third layers are more ordered than the first layer. Could the author explain the possible reason? As widely reported in previous works (e.g. *ACS Nano* 2019, 13, 9270–9278), halogen adatoms on Au(111) may drive the ordering of the chain-like structures through hydrogen bonding interactions. Do the authors think a similar mechanism works in your system that iodine adatoms promote the ordering of 5-AGNRs?
10. Why was an iodine-covered Au(111) substrate instead of a bare Au(111) chosen for the investigation of multilayer-growth of 5-AGNRs?
11. The applications of electrochemically produced GNRs should be included in the conclusion section.

Response to reviewer comments

Reviewer #1 (Remarks to the Author):

In this work, the authors present an electrochemical synthesis/growth of graphene nanoribbons (GNRs) on substrates using the naphthalene compound substituted with an electron-rich butoxy group. The redox behavior of this precursor compound is demonstrated to be responsible for the so-called heterochiral polymerization involving dicationic mechanism followed by the dehydrogenation under the electrochemical condition with voltage. As a result, the authors show the formation of N=5 GNR with edge substitution on the surface with layer-by-layer deposition. Moreover, the authors demonstrate the functionality of the resultant GNRs. Overall, the proposed electrochemical synthesis method for the GNR involving novel precursors and mechanisms is quite interesting and can broaden the current synthetic scopes for the GNR chemistry. Nevertheless, the following comments shall be carefully addressed before further consideration of the current work:

Thank you very much for your insightful comments.

Major comments:

Q1: what is the exact role of the substrate during the electrochemical synthesis in this work? The authors used the Au (111), I₂ passivated Au, and ITO substrates, respectively. The authors also claimed the crucial role of the electrical double layer (EDL) at the liquid-solid interface, which has a thickness of about 1 nm. This also raises the question of how thick (how many layers) of GNRs can be grown with the EDL confinement with this technique?

A1: We used three substrates as working electrodes for different purposes. The first two, Au(111) and iodine-covered Au(111) substrates, examined the GNR structures using STM measurements. We employed a transparent conductive ITO substrate for optical measurements.

Sakaguchi *et al.* (*Nat. Mater.* **3**, 551–557 2004) synthesized polythiophene on a gold substrate and observed polymer chains using STM. They also found that iodine-covered Au(111) substrates improved lattice matching between polythiophenes and substrates, resulting in improvement of polymer orientations. Our study found that electrochemically produced GNRs could be synthesized on both Au(111) and iodine-covered Au(111) substrates. However, we observed that the orientation of GNRs improved when

synthesized on an iodine-covered Au(111) substrate. This improvement is due to lattice matching between GNRs and the substrate.

We added the following sentences to the main text and modified Supplementary Fig. 2.

Page 4, Line 4-8

“Atomic flat Au(111) and iodine-covered Au(111) substrates were utilized for scanning tunneling microscopy (STM) measurements. Iodine-covered Au(111) improves the alignment of GNRs due to lattice matching between GNRs and the iodine atoms on the substrate (Supplementary Figs. 1 and 2). A transparent ITO substrate was used for optical measurements.”

Page 4, Line 26-31

“Lattice parameters of iodine-covered Au(111) in a compressed hexagonal structure, along the a, b and c axes are 4.2, 5.4 and 4.2 Å, respectively, while those of Au(111) are 2.9 Å. The periodicity of GNR (8.5 Å) mismatches with the atom spacing of substrates (Au(111) and iodine-covered Au(111)) by 3.53% and 1.17%, respectively. This indicates that iodine-covered Au(111) is more suitable than Au(111) for epitaxial GNR growth.”

Supplementary Fig. 2. (a) LT-STM image of iodine-covered Au (111) with a superimposed structure of GNR. The inset shows a, b, and c axes, and a compressed hexagonal lattice structure. I-I distances along the a, b, and c axes are 4.2, 5.4, and 4.2 Å, respectively. The mismatch between GNR and substrate is 1.17%. (b) Lattice matching between the GNR and Au(111). Au-Au distances along the a, b, and c axes are 2.9, 2.9, 2.9 Å, respectively. The mismatch between GNR and substrate is 3.53%.

For the second issue regarding the thickness of GNRs, STM cannot characterize the

thickness of GNR films because the substrates are completely covered by GNRs. Therefore, AFM measurements were carried out to estimate the thickness of multilayered GNRs transferred from the Au substrate to the Si substrate. For this purpose, we prepared 5 different GNR samples grown on an iodine-covered Au(111) by applying voltage pulses of 3, 4, 10, 20, and 30 at 5 V for 0.5 seconds, which were transferred to Si substrates.

We added the following sentences to the text.

Page 9, Line 21-28

“In order to investigate the thickness of multilayered GNRs, AFM measurement of transferred GNRs from iodine-covered Au(111) to Si substrate was carried out because STM cannot characterize the thickness of GNR films due to substrates fully covered by GNRs (Supplementary Figs. 21 and 22). It was observed that the thickness and Raman intensity of GNR films were directly proportional to the number of voltage pulses applied. As per the STM characterization, the estimated thickness of a monolayer was 0.114 nm (Fig. 2c). When a voltage pulse of 5 V was applied, a 2.46-layer was formed as per the AFM characterization (Fig. 4e and Supplementary Fig. 22).”

Page 16, Line 4-10

“Atomic Force Microscope (AFM) Characterization

We prepared samples to determine the thickness of GNRs using electrochemical means on an iodine-covered Au(111) by applying voltage pulses of 3, 4, 10, 20, and 30 at 5 V for 0.5 seconds. The GNRs were then transferred to the Si substrate using the transfer method. To investigate the thickness of GNRs on the Si substrate, we performed AFM measurements (CSI, AFM Galaxy Dual Controller and Digital Instruments, MultiMode AFM-2) (Supplementary Fig. 22).”

We reorganized Fig. 2 and included a height indicator of GNR thickness in Fig. 2(c). Additionally, we added Fig. 4(e) and Supplementary Fig. 22.

Fig. 2. Characterization of electrochemically produced GNRs. (a) LT-STM image (-0.97 V, 580 pA) of electrochemically produced sample (5 V, 0.5 sec, 3 cycles). Red and blue dotted lines indicate the marks for (b). (b) Magnified LT-STM image of the strand with the structure of a GNR. (c) Cross-sectional analysis of strand periodicity (top) and width (bottom). (d) STS profile and simulated DFT bandgap. The inset shows the measured location and the simulated bandgap value. (e) Optical absorption spectrum of GNR films on ITO (black) and the simulation (red). The inset displays a photograph of GNR films. (f) Experimental (top) and simulated (bottom) Raman spectra. (g) MALDI-FT-ICR MS spectrum of GNRs showing the number of units. (h) Cyclic voltammetry of GNR films on the Au(111) (red) and TTF (black) in *o*-DCB electrolyte solution. The inset shows chemical structures.

Fig. 4. Voltage cycle dependence on electrochemical GNR growth. LT-STM images of electrochemically produced samples (5 V, 0.5 sec) with (a) one cycle (STM measurement conditions: -1.95 V, 160 pA), (b) two cycles (-0.31 V, 580 pA), (c) three cycles (-0.76 V, 580 pA) and (d) four cycles (-0.81 V, 580 pA), respectively. (e) Thickness and Raman intensity of electrochemically produced GNR films transferred from iodine-covered Au(111) to Si substrates with different numbers of cycles (5 V, 0.5 sec). The inset shows an AFM image of a GNR film (30 cycles) on Si.

Supplementary Fig. 22. (a-e) AFM images, cross sections and Raman spectra of GNRs obtained after applying various numbers of voltage cycles (3, 4, 10, 20, and 30 cycles) (5 V, 0.5 sec).

Q2: what is the influence of temperature on the GNR growth in this work? The authors performed the experiments at 80°C. Does the temperature mainly influence the polymerization step, or the dehydrogenation step, or both? The GNRs grown on the substrates are generally too short (< 10 nm, and it is not clear why I2-covered Au substrate gives a longer ribbon); any specific reason for this? This would refer to the polymerization step. Then what is the critical role of achieving a high polymerization degree with the precursor monomer and electrochemical method?

A2: Based on the newly added Supplementary Fig. 3, the STM image shows that at room temperature, GNRs require at least 30 more cycles of voltage pulses to fully cover the sample compared to the sample that was grown at 80°C with only 3 cycles of voltage pulses. This fact indicates that higher temperatures enhance GNR growth, making temperature crucial for electrochemical growth of GNRs. According to Çiftçi *et al. Polym. Bull.* **66**, 747–760 (2011), higher temperatures enhance polymerization by overcoming activation energy and it is believed that the dehydrogenation reaction is supposed to be same. An 80°C reaction temperature also enhances diffusion of the precursor, which improves mass transfer in the electric double layer (EDL) (M. Petrowsky *et al. J. Phys. Chem. B* **113**, 5996–6000 (2009)).

We added the the following explanation to the text.

Page 4, Line 8-10

“The reaction temperature of 80°C enhances formation of GNRs by overcoming activation energy and improving precursor diffusion, compared to room temperature (Supplementary Fig. 3)^{41,42}.”

We added the STM image of GNR synthesized at room temperature as Supplementary Fig. 3.

Supplementary Fig. 3. Ambient STM image of electrochemically produced GNR on iodine-covered Au(111) at room temperature (5 V, 0.5 sec, 30 cycles).

Regarding the second issue about the critical role of achieving a high degree of polymerization, controlling the nucleation rate on the surface is critical to producing longer GNRs. If the density of the precursors adsorbed on the substrate is too high, the grown GNRs are likely to collide with each other, leading to a decrease in chain length. This suggests that the precursor density on the substrate should be lower to obtain longer chains. In future, we plan to conduct experiments to modify substrates not only with iodine but also with different atoms or molecules to control the interaction between precursors and substrates.

Q3: It would be valuable if the authors could briefly discuss the polymerization mechanism with the proposed dication polymerization. Is this the chain-growth polymerization or step-growth polymerization? The different mechanisms also influence the homogeneity of the GNRs.

A3: Based on our research, we have found that polymerization reaction is likely to occur at high electron density sites on dication species that are adsorbed on the electrode under

a high electric field in the electric double layer (EDL). These high electron density sites in dications then react with neutral precursors that approach from outside the EDL. We believe that the growth mode of GNRs follows a chain-growth mechanism since the di-cation active site located at the growth terminal cannot diffuse on the surface but instead firmly binds to it.

We added the following text.

Page 9, Line 3-5

“The GNR growth mode is believed to be chain-growth because the di-cation active site at the growth terminal is not able to diffuse on the surface but bind tightly to it.”

Q4: in page 4, the authors obtained an optical bandgap of 1.0 eV for the grown GNRs on the ITO substrate and claimed that this value agrees with the result from STS measurement (0.84 eV on the iodine-covered Au (111)). Nevertheless, these two values are still quite deviated from each other. Why? Can the latter be due to the coupling between the GNRs and the Au substrate?

A4: We found that the peak at ~1400 nm is due to an equipment artifact which prevented accurate estimation. Therefore, we removed this peak from the absorption spectrum of the GNR film and reconsidered the Tauc plot. As a result, the optical band gap was estimated at 0.85 eV. This result is consistent with that of 0.84 eV obtained from STS measurement.

We modified the text as follows.

Page 4, Line 41-43

“The bandgap of 0.85 eV obtained from the Tauc plot in the absorption spectrum agrees with that obtained from STS (Fig. 2d and Supplementary Fig. 7)”

We modified Fig. 2(e) and Supplementary Fig. 7.

Fig. 2. Characterization of electrochemically produced GNRs. (a) LT-STM image (-0.97 V, 580 pA) of electrochemically produced sample (5 V, 0.5 sec, 3 cycles). Red and blue dotted lines indicate the marks for (b). (b) Magnified LT-STM image of the strand with the structure of a GNR. (c) Cross-sectional analysis of strand periodicity (top) and width (bottom). (d) STS profile and simulated DFT bandgap. The inset shows the measured location and the simulated bandgap value. (e) Optical absorption spectrum of GNR films on ITO (black) and the simulation (red). The inset displays a photograph of GNR films. (f) Experimental (top) and simulated (bottom) Raman spectra. (g) MALDI-FT-ICR MS spectrum of GNRs showing the number of units. (h) Cyclic voltammetry of GNR films on the Au(111) (red) and TTF (black) in *o*-DCB electrolyte solution. The inset shows chemical structures.

Supplementary Fig. 7. Tauc plot obtained from the optical absorption spectrum of electrochemically produced GNR film on an ITO electrode showing the bandgap value.

Q5: For the ITO substrate-grown GNRs, the exact structural proof is somehow lacking (only UV and Raman). Did the authors compare these results obtained for both ITO-substrate and Au-substrate grown samples? From Figure 2d, it is not clear to me why there is a strong band at around 1400 nm absorption.

A5: On ITO, roughness of the substrate prevents STM measurements of products. Therefore, we can only characterize those using Raman spectra. As shown in newly added Supplementary Fig. 10, we compared Raman spectra between the GNR formed on the ITO and that on an Au substrate. The results match well.

We have also added the following sentences to the text.

Page 5, Line 12-14

“Roughness of the substrate prevents STM measurements of the products on ITO. (Supplementary Fig. 9). Products are confirmed as GNRs due to their Raman spectrum which correspond to those of GNRs produced on Au(111) (Supplementary Fig. 10).”

We added Supplementary Figs. 9 and 10.

Supplementary Fig. 9. AFM image and cross section of the ITO substrate.

Supplementary Fig. 10. Raman spectra of electrochemically produced GNRs formed on iodine-covered Au(111) and ITO substrates (5 V, 0.5 sec, 30 cycles).

For the next issue about strong band at around 1400 nm absorption, we already answered in Q4 with the addition of Fig. 2(e) and the modification of Supplementary Fig. 7.

Q6: Figures 4 and S3 show that many GNRs appear kinked (rather than straight); why? Could it be due to the missed/non-selective coupling reaction of precursor monomers during the polymerization step?

A6: We recognize the importance of discussing kinks and added the following sentences to the text.

Page 9, Line 16-20

“LT-STM images (Fig. 4c and d) show that the majority of the strands are composed of linear chains that are coupled in a regioselective manner. However, kinks are sometimes observed at the ends of these straight chains, suggesting that there may have been collisions with neighbouring chains during their growth. This collision indicates that coupling is non-regioselective.”

Q7: For the photoconductive property study of GNRs on the substrate, why was FeCl₃ doping employed? How about the device performance without FeCl₃ doping?

A7: Since GNRs are strong electron-donating materials, FeCl₃ doping was used as an electron acceptor to generate holes in the GNRs by photoirradiation. This resulted in the

enhancement of photoconductivity. (B. Yurash *et al. Nat. Mater.* **18**, 1327–1334 (2019); J. Min *et al. Adv. Funct. Mater.* **33**, 2212825 (2023); E. Poverenov *et al. J. Am. Chem. Soc.* **136**, 5138–5149 (2014)).

According to modified Supplementary Figure 30, it can be concluded that undoped GNRs do not exhibit photoconductivity.

We added these sentences to the text and modified Supplementary Fig. 30.

Page 12, Line 21-Page 13, Line 1

“Additionally, we confirmed that electrochemically produced GNRs without FeCl₃ doping did not show photoconductivity (Supplementary Fig. 30). These results indicate efficient generation of photo-carriers in GNRs electrochemically produced with electron transfer from photoexcited states of GNRs to FeCl₃, which acts as an electron-acceptor, resulting in hole transport.”

Supplementary Fig. 30. Photoconductive properties of FeCl₃-doped 5-AGNR and 7-AGNR and undoped electrochemically produced GNR.

Minor comments:

Q1: in page 6, the authors claimed the “GNRs demonstrate exceptional chemical stability”. It is not clear to me how the chemical stability was evaluated in this work.

A1: According to your suggestion, we modified this sentence.

Page 6, Line 14-15

“Electrochemically produced GNRs demonstrate excellent electrochemical cycling durability.”

Q2: it would be better to call the “electron-donor” to “electron-rich” in the context for the GNRs?

A2: We greatly appreciate this comment. However, we would like to clarify our decision regarding the use of certain terms. We conducted a search for the phrases "electron-donating conjugated polymers" and "electron-rich conjugated polymers" and found 1710 and 1840 hits, respectively. In contrast, our search yielded 1.17 million hits for "electron-withdrawing conjugated polymers" and 390 hits for "electron-poor conjugated polymers". Based on these findings, we determined that the best terms are "electron-donating" and "electron-withdrawing." After careful consideration, we decided to use "electron-donating" in our manuscript.

Q3: do I understand it correctly that the dication is fully distributed (rather than being located on certain carbon atoms) on the naphthalene compound in Figure 3d? Of course, this makes it difficult to judge which position of carbon atom in naphthalene would be more favorable for the reaction. The authors shall also label the positions “1 and 5” in the Figure 3d.

A3: Our electron density mapping simulation revealed that the carbon atom at the 1-position has a localized hole density, while the other positions have a delocalized density.

We added the following sentences.

Page 7, Line 24- Page 8, Line 3

“The DFT-calculated electrostatic map indicates that the carbon at position 1 in the dication has the highest hole density, suggesting that it is an active site, whereas other positions have delocalized densities (Fig. 3d and 1b).”

According to your suggestion, we improved Fig. 3(d).

Fig. 3. Mechanism of electrochemical growth of GNR. (a) Electro-absorption spectrum after voltage application (3 V, 15 sec; top) and simulated spectra of di-cation and cation radicals (bottom). The inset shows the experimental setup. (b) The electrochemical voltage dependence on the absorption of a GNR film formed on ITO at 650 nm. The inset shows the setup and voltage-dependent absorption spectra of a GNR film. (c) The correlation between the di-cation absorption at 1600 nm in the cell and GNR absorption at 650 nm in the films. (d) Spin density of the cation radical (left), electron density of precursor (center) and di-cation (right). (e) Reaction mechanism of GNR growth. (f) The unit number dependence of total energies of the heterochiral-coupled GNR (red) and semi-periodical-coupled GNR (black). The inset shows structures. (g) Energy levels of HOMO-1 and LUMO for possible trimer products. The inset shows examples of products. The dotted red circle indicates the highest HOMO-1.

Q4: The quality and readability of Figure S13 shall be improved.

A4: As shown in the modified Supplementary Fig. 17 (former Supplementary Fig. 13), we enlarged the size of the figure and font to improve readability.

Supplementary Fig. 17. (a) Possible reaction pathways and products of the tetramer. (b) Total energies of possible tetramer products. The red square shows minimum energy.

Q5: the two important review papers on the GNRs shall be cited (Chem. Soc. Rev. 2015, 44, 6616; Acc. Chem. Res. 2022, 55, 23, 3322).

A5: We added the following as References 13 and 14.

- “13. Narita, A., Wang, X.-Y., Feng, X. & Müllen, K. New advances in nanographene chemistry. *Chem. Soc. Rev.* **44**, 6616–6643 (2015).
 14. Niu, W., Ma, J. & Feng, X. Precise Structural Regulation and Band-Gap Engineering of Curved Graphene Nanoribbons. *Acc. Chem. Res.* **55**, 3322–3333 (2022).”

Reviewer #2 (Remarks to the Author):

In the manuscript "Electrochemical on-surface synthesis of a strong electron-donating graphene nanoribbon catalyst" Hiroshi Sakaguchi and co-workers present an interesting electrochemical approach for the synthesis of graphene nanoribbons equipped with side groups. The authors show the synthesis of well-shaped GNRs containing side butoxy substituents. I believe that the achievement is interesting and appealing for a broad readership. Indeed the on-surface approach suffers from the inability of multilayer generation and lack of resistance of a range of substituents to high temperature treatment needed for reaction initiation. Presented approach provides an interesting step forward. I recommend the manuscript for publication in Nature Communications after minor revision addressing the following points:

Thank you very much for your insightful comments.

Q1: the authors state (e.g. in abstract) that they successfully demonstrate highly organized GNR, I am not convinced that one could describe the system as "highly organized".

A1: According to your suggestion, we deleted the expression, "highly organized".

Page 1, line 18-20

“We successfully demonstrate layer-by-layer growth of ~~highly organized~~ strong electron-donating GNRs on electrodes at temperatures $<80^{\circ}\text{C}$ without decomposing functional groups.”

Q2: the manuscript needs some language corrections, whereas in general it is clearly written some sentences are difficult to follow, below I attach a few examples - in abstract "Electron-donating GNRs exhibiting one of the strongest electron-donating properties known, enable extraordinary performance on silicon-etching catalytic activities better than noble metals, with superior photoconductive properties.", page 2 : "The redox potential of electrochemically produced GNRs shows one of the strongest electron-donating substances discovered.", page 9: "This result suggests that the oligomers of GNRs may cover the electrode." The above mentioned sentences are unclear. I suggest careful reading and rephrasing especially too long sentences.

A2: We have improved these phrases.

Page 1, Line 21-24

“Electrochemically produced GNRs exhibiting one of the strongest electron-donating properties known, enable extraordinary silicon-etching catalytic activity, exceeding those of noble metals, with superior photoconductive properties.”

Page 2, Line 39-41

“Electrochemically produced GNRs have a lower oxidation potential compared to tetrathiafulvalene (TTF) indicating strong electron-donating properties.”

We removed the sentence, “This result suggests that the oligomers of GNRs may cover the electrode”, as it seems duplicated. We also added a sentence about the coverage rate.

Page 9, Line 9-11

“When a single voltage pulse of 5 V is applied for 0.5 seconds to the electrode, the resulting LT-STM image displays a random arrangement of short chains, all less than 1 nm in length (Fig. 4a). ~~This result suggests that the oligomers of GNRs may cover the electrode.~~”

Q3: what is the coverage (estimation) in Fig.4a?

A3: We added the sentence below.

Page 9, line 11-12

“The coverage ratio of this sample is estimated to be about 82.3%.”

Q4: were the samples degassed after transferring to UHV?

A4: We added the sentence below to the Methods.

Page 15, line 1-2

“All samples were degassed in UHV with annealing at 150°C for 1 h before STM measurements.”

Q5: refs 50-51 appear under data availability.

A5: The section on data availability was relocated and modified. References 60 and 61 (former 50 and 51) were moved to supplementary information.

Page 17, line 25-26

“Data Availability

The data presented in this study are available on request from the corresponding author.”

~~60. Clark, S. J. *et al.* First principles methods using CASTEP. *Zeitschrift für Kristallographie—Crystalline Materials* **220**, 567–570 (2005).~~

~~61. Perdew, J. P., Burke, K. & Ernzerhof, M. Generalized Gradient Approximation Made Simple. *Phys. Rev. Lett.* **77**, 3865–3868 (1996).~~

Q6.- in page 4 the authors say "Additionally, the Raman spectrum of electrochemically produced GNRs shows the G-bands (1600 cm⁻¹), D-bands (1000-1400 cm⁻¹), and radial breathing-like mode bands (544 cm⁻¹) (Fig. 2e). These peaks match those obtained from DFT simulation based on our GNR model", whereas in my opinion the experimental data shown in Figure 2e differs quite substantially from the calculated one. I would suggest to discuss the issue in more detail.

A6: As shown in Figure 2(f), we recalculated the Raman spectrum by Gaussian module with 10 repeats of GNR. The calculated spectrum fits the experimental spectrum better.

We added the following material to the text.

Page 5, Line 2-7

“A simulation of Raman was conducted using a Gaussian program on a 10-mer of GNR in a gas-phase. Peaks of G-bands and radial breathing-like mode bands matched the experimental results. However, in comparison to experimental data, the D-band simulation covers the experimental peak positions, whereas the intensity in the simulation deviates from the experimental. This could be due to the dispersion of GNRs with varying lengths.”

We modified Figure 2(f).

Fig. 2. Characterization of electrochemically produced GNRs. (a) LT-STM image (-0.97 V, 580 pA) of electrochemically produced sample (5 V, 0.5 sec, 3 cycles). Red and blue dotted lines indicate the marks for (b). (b) Magnified LT-STM image of the strand with the structure of a GNR. (c) Cross-sectional analysis of strand periodicity (top) and width (bottom). (d) STS profile and simulated DFT bandgap. The inset shows the measured location and the simulated bandgap value. (e) Optical absorption spectrum of GNR films on ITO (black) and the simulation (red). The inset displays a photograph of GNR films. (f) Experimental (top) and simulated (bottom) Raman spectra. (g) MALDI-FT-ICR MS spectrum of GNRs showing the number of units. (h) Cyclic voltammetry of GNR films on the Au(111) (red) and TTF (black) in *o*-DCB electrolyte solution. The inset shows chemical structures.

Q7: the data shown in Figure 4 are acquired with very different STM settings (mainly voltage), why?

A7: This was due to a technical matter. In the case of Fig 4(a), which is not fully covered by GNRs, a clear image was obtained at only high voltage (-1.95 V). This may have been caused by the conductivity difference between GNR and Au substrates.

Q8: can the authors estimate the number of layers in Fig. 4d?

A8: STM cannot characterize the thickness of GNR films because the substrates are completely covered by GNRs. Therefore, AFM measurements were carried out to estimate the thickness of multilayered GNRs transferred from the iodine-covered Au(111)

substrate to the Si substrate. For this purpose, we prepared 5 different GNR samples grown on an iodine-covered Au(111) by applying voltage pulses of 3, 4, 10, 20, and 30 at 5 V for 0.5 seconds, which were transferred to Si substrates.

We added the following material to the text.

Page 9, Line 21-28

“In order to investigate the thickness of multilayered GNRs, AFM measurement of transferred GNRs from iodine-covered Au(111) to Si substrate was carried out because STM cannot characterize the thickness of GNR films due to substrates fully covered by GNRs (Supplementary Figs. 21 and 22). It was observed that the thickness and Raman intensity of GNR films were directly proportional to the number of voltage pulses applied. As per the STM characterization, the estimated thickness of a monolayer was 0.114 nm (Fig. 2c). When a voltage pulse of 5 V was applied, a 2.46-layer was formed as per the AFM characterization (Fig. 4e and Supplementary Fig. 22).”

Page 16, Line 4-10

“Atomic Force Microscope (AFM) Characterization

We prepared samples to determine the thickness of GNRs using electrochemical means on an iodine-covered Au(111) by applying voltage pulses of 3, 4, 10, 20, and 30 at 5 V for 0.5 seconds. The GNRs were then transferred to the Si substrate using the transfer method. To investigate the thickness of GNRs on the Si substrate, we performed AFM measurements (CSI, AFM Galaxy Dual Controller and Digital Instruments, MultiMode AFM-2) (Supplementary Fig. 22). ”

We added a height indicator of thickness of GNR monolayer to Fig. 2(c). We also added Fig. 4(e) and Supplementary Fig. 22.

Fig. 2. Characterization of electrochemically produced GNRs. (a) LT-STM image (-0.97 V, 580 pA) of electrochemically produced sample (5 V, 0.5 sec, 3 cycles). Red and blue dotted lines indicate the marks for (b). (b) Magnified LT-STM image of the strand with the structure of a GNR. (c) Cross-sectional analysis of strand periodicity (top) and width (bottom). (d) STS profile and simulated DFT bandgap. The inset shows the measured location and the simulated bandgap value. (e) Optical absorption spectrum of GNR films on ITO (black) and the simulation (red). The inset displays a photograph of GNR films. (f) Experimental (top) and simulated (bottom) Raman spectra. (g) MALDI-FT-ICR MS spectrum of GNRs showing the number of units. (h) Cyclic voltammetry of GNR films on the Au(111) (red) and TTF (black) in *o*-DCB electrolyte solution. The inset shows chemical structures.

Fig. 4. Voltage cycle dependence on electrochemical GNR growth. LT-STM images of electrochemically produced samples (5 V, 0.5 sec) with (a) one cycle (STM measurement conditions: -1.95 V, 160 pA), (b) two cycles (-0.31 V, 580 pA), (c) three cycles (-0.76 V, 580 pA) and (d) four cycles (-0.81 V, 580 pA), respectively. (e) Thickness and Raman intensity of electrochemically produced GNR films transferred from iodine-covered Au(111) to Si substrates with different numbers of cycles (5 V, 0.5 sec). The inset shows an AFM image of a GNR film (30 cycles) on Si.

Supplementary Fig. 22. (a-e) AFM images, cross sections and Raman spectra of GNRs obtained after applying various numbers of voltage cycles (3, 4, 10, 20, and 30 cycles) (5 V, 0.5 sec).

Q9: the etching approach seems very interesting, do the authors know the limit of patches size that could be still applied for etching (the smallest patches)?

A9: We added this sentence to the text and to Supplementary Fig. 25.

Page 11, Line 29

“The smallest etched area was $1 \times 2 \mu\text{m}^2$ (Supplementary Fig. 25).”

Supplementary Fig. 25. (a) Optical microscope image of GNR-covered Si before etching. (b) SEM image after etching. (c) Magnified SEM image of the red circle in (b).

Q10: would the precursor molecules also provide etching properties (without formation of GNRs)?

A10: We conducted an etching experiment on a Si substrate that was covered with a precursor. We used the same conditions as the electrochemically produced GNR-catalyzed Si etching experiment. As shown in Supplementary Fig. 26, the precursor did not exhibit any etching catalytic activity.

We added the following explanation to the text and to Supplementary Fig. 26.

Page 11, Line 32-33

“We also confirmed that precursor did not etch Si substrate. (Supplementary Fig. 26).”

Supplementary Fig. 26. SEM images of top view (a) and cross-sectional view (b) of the 2-butoxynaphthalene precursor-covered Si after etching.

Q11: the patches used for etching seem to be irregular, is there any idea to prepare them in a regular manner with desired dimensions?

A11: We conducted a preliminary study on the phenomenon of the Si etching catalyst. As you suggested, the unique nature of GNRs will enable creation of Si nanoarchitecture. In the future, we plan to perform nanoscale Si etching using patterned GNR dots.

Q12: the authors discuss the mechanism of electrochemical synthesis of GNRs, could the approach be easily transferred for the synthesis of e.g. wider GNRs?

A12: We believe that an electrochemical on-surface synthetic approach can produce various types of GNRs. To achieve wider GNRs, new precursors capable of generating cationic species of precursors by applying high electric fields may be required to allow for dicationic polymerization. DFT calculations to predict the electron density may assist in designing suitable precursors with appropriate electronic structures.

We added this sentence to the text.

Page 13, Line 22-23

“Our electrochemical on-surface synthesis can be applied to various aromatic precursors, leading to wider GNRs.”

Reviewer #3 (Remarks to the Author):

In the manuscript, Sakaguchi et al. reported a novel electrochemical on-surface synthesis approach and obtained multilayered 5-AGNRs which are doped by electron-donating groups. The conventional on-surface synthesis of GNRs often needs a high-temperature annealing and could typically get only monolayered GNR products. The electrochemical method reported here thus provides an exciting pathway for the large-scale fabrication of high-quality GNRs, which could be used for the real applications in nanodevices, as also demonstrated by the authors. The data is of high quality and the majority of conclusion are well supported by a series of control experiments and DFT calculations. Therefore, I would recommend its publication in Nat. Commun. once a few technical points and drawbacks are satisfactorily addressed.

Thank you very much for your insightful comments.

Q1: The paper is a little bit too long. I suggest the authors shorten some less important sections to make the manuscript more concise, which will also make the readers easier to grasp the main meaning of the paper.

A1: Thank you for this suggestion. The main text is currently 3839 words, which is well below the word limit of 5000, excluding Abstract, Methods, References and Figure legends.

Q2: In the introduction section and also some positions of the main text, the authors stated that on-surface synthesis is limited to monolayer growth. This is actually not always true, e.g. J. Am. Chem. Soc. 2023, 145, 10126-10135.

A2: We modified the text and cited the paper you recommended (Ref. 45).

Page 9, Line 32-37

“One exception, involving bilayered GNRs was reported using UHV on-surface synthesis⁴⁵. This method accomplishes the dehydrogenation reaction of prepolymers, originating from a domino-like effect remotely triggered by the direct contact between the GNR segment and the gold substrate. Direct contact between the prepolymer and a metal substrate is necessary for second-layer growth, which still remains a limitation of multilayer growth.”

References

“45. Ma, C. *et al.* Remote-Triggered Domino-like Cyclodehydrogenation in Second-Layers Topological Graphene Nanoribbons. *J. Am. Chem. Soc.* **145**, 10126–10135 (2023).“

Q3: There are more examples about edge-functionalization of GNRs via on-surface synthesis, such as ketone and amino functionalization, e.g. *Nat. Chem.* 2022, 14, 1451-1458; *Nano Lett.* 2022, 22, 164-171; *ACS Nano* 2020, 14, 1895-1901. The ketone groups could even possibly bring in magnetism to a GNR system. The author should consider to cite these works.

A3: According to your suggestion. we cited these works.

References

“19. Lawrence, J. *et al.* Circumventing the stability problems of graphene nanoribbon zigzag edges. *Nat. Chem.* **14**, 1451–1458 (2022).

20. Li, J. *et al.* Band Depopulation of Graphene Nanoribbons Induced by Chemical Gating with Amino Groups. *ACS Nano* **14**, 1895–1901 (2020).

21. Wang, T. *et al.* Magnetic Interactions Between Radical Pairs in Chiral Graphene Nanoribbons. *Nano Lett.* **22**, 164–171 (2022).”

Q4: The dI/dV STS shown in Fig. 2c is not convincing. The authors should measure a spectrum on a bare Au(111) surface using the same tip to compare. This would help one to know that the resonances in STS are really contributed by the GNR or simply from the tip itself. If a STS on a bare Au(111) is not possible to measure because the surface is fully covered by GNR, the authors should try to map out the resonances in dI/dV spectrum, i.e. dI/dV maps at ~ -0.2 and $+0.6$ V to make sure they are “real” conductance resonances of GNRs.

A4: We were unable to take STS measurements with an Au(111) surface using the same tip for comparison because the sample was completely covered with GNRs. Instead, we measured dI/dV mapping of GNRs at -0.2 (HOMO) and $+0.6$ V (LUMO). We also calculated the LDOS using the DMol3 module in Materials Studio. The experimental

dI/dV mapping roughly agrees with the simulation result, as shown in Supplementary Fig. 6.

We added the following sentences to the text and to the Methods.

Page 4, Line 38-39

“Furthermore, we performed dI/dV mapping of GNR. The experimental dI/dV mapping agrees roughly with the simulation result. (Supplementary Fig.6). “

Page 15, Line 6-8

“The differential conductance map was recorded with set V_{sample} set at -0.2 V or 0.6 V and I at 80 pA by a digital lock-in amplifier with a modulation frequency of 350 Hz and an AC voltage of 2 mV.”

We added Supplementary Fig. 6.

Supplementary Fig. 6. Experimental LT-STM topography of electrochemically produced GNR on iodine-covered Au(111) (a) and molecular models (b). Experimental constant-height dI/dV maps (c,e) and simulation of LDOS maps (d,f).

Q5: How did the authors perform DFT calculations for the simulation of a bandgap? In gas phase or on a surface? The result from the former cannot be directly compared to the experimental values because the electron screening effect, charge transfer, etc, are not

considered. The difference could be very large.

A5: We agree that there is a difference between the bandgap obtained from experimental data and that from DFT simulation. This difference is due to factors such as the neglect of electron screening and charge transfer. We have taken your comment into consideration and have made changes to the text accordingly. Instead of comparing the experimental bandgap obtained from STS and that from theory, we validated the STS experimental value by comparing it using other experimental methods, such as optical absorption. The band gap value obtained from STS (0.84 eV) is consistent with an optical bandgap of 0.85 eV. Therefore, we conclude that the experimental bandgap obtained from STS is reasonable.

We added the following explanation to the text.

Page 4, Line 32-38

“Scanning tunneling spectroscopy (STS) of electrochemically produced GNRs on iodine-covered Au(111) reveals a bandgap of 0.84 eV, while density functional theory (DFT) provides a value of 0.74 eV (Fig. 2d). This minor difference between experimental results and DFT calculations, may be attributed to the excluded intermolecular interactions. The bandgap measured by STS is consistent with that measured by optical means, which we will discuss later.”

Q6: Fig. 2b is too small. A good match between the molecular model and the corresponding STM image is the most important evidence to confirm the formation of the 5-AGNR functionalized by OBU groups. Please make it larger.

A6: According to your suggestion, we made the figure larger.

Fig. 2. Characterization of electrochemically produced GNRs. (a) LT-STM image (-0.97 V, 580 pA) of electrochemically produced sample (5 V, 0.5 sec, 3 cycles). Red and blue dotted lines indicate the marks for (b). (b) Magnified LT-STM image of the strand with the structure of a GNR. (c) Cross-sectional analysis of strand periodicity (top) and width (bottom). (d) STS profile and simulated DFT bandgap. The inset shows the measured location and the simulated bandgap value. (e) Optical absorption spectrum of GNR films on ITO (black) and the simulation (red). The inset displays a photograph of GNR films. (f) Experimental (top) and simulated (bottom) Raman spectra. (g) MALDI-FT-ICR MS spectrum of GNRs showing the number of units. (h) Cyclic voltammetry of GNR films on the Au(111) (red) and TTF (black) in *o*-DCB electrolyte solution. The inset shows chemical structures.

Q7. Fig. 3: Please mark the position of site 1 and 5 near the molecular structure. Readers with a poor chemistry background cannot identify the numbers of carbon sites easily.

A7: To facilitate reader understanding, we added the position of all sites near the molecular structure, as shown in Figure 3 (d).

Fig. 3. Mechanism of electrochemical growth of GNR. (a) Electro-absorption spectrum after voltage application (3 V, 15 sec; top) and simulated spectra of dication and cation radicals (bottom). The inset shows the experimental setup. (b) The electrochemical voltage dependence on the absorption of a GNR film formed on ITO at 650 nm. The inset shows the setup and voltage-dependent absorption spectra of a GNR film. (c) The correlation between the di-cation absorption at 1600 nm in the cell and GNR absorption at 650 nm in the films. (d) Spin density of the cation radical (left), electron density of precursor (center) and di-cation (right). (e) Reaction mechanism of GNR growth. (f) The unit number dependence of total energies of the heterochiral-coupled GNR (red) and semi-periodical-coupled GNR (black). The inset shows structures. (g) Energy levels of HOMO-1 and LUMO for possible trimer products. The inset shows examples of products. The dotted red circle indicates the highest HOMO-1.

Q8: Page 8. “According to the STM measurements (Supplementary Fig. 15), these symmetric precursors are unable to produce GNRs under the same electrochemical conditions using 2-butoxynaphthalene.” These STM images were obtained at ambient pressure. Therefore, it is reasonable to argue that the absence of GNR is simply due to the low resolution or a low coverage.

A8: To confirm this, we re-measured samples by LT-STM to obtain images with higher resolution, and improved the quality of Supplementary Fig. 19.

Supplementary Fig. 19. LT-STM (0.2 V, 5 pA) images and Raman spectra of electrochemically treated Au (111) in **a** 5 mM sample solution of (a, c) 2,6-dibutoxynaphthalene, (b, d) 2,7-dibutoxynaphthalene with 0.1 M of TBAPF₆ in *o*-DCB after voltage application (5 V, 0.5 s, 3 cycles), respectively. **Panels** of (c) and (d) show chemical structures.

Q9: As for the iodine-covered Au(111), the authors demonstrated that the GNR in the second and third layers are more ordered than the first layer. Could the author explain the possible reason? As widely reported in previous works (e.g. ACS Nano 2019, 13, 9270–9278), halogen adatoms on Au(111) may drive the ordering of the chain-like structures though hydrogen bonding interactions. Do the authors think a similar mechanism works in your system that iodine adatoms promote the ordering of 5-AGNRs?

A9: As per your comment, we added the following text and the references.

Page 11, Line 3-9

“ It has been reported that the ordering of poly-*p*-phenylene (PPP) is driven by surface-bound bromine⁴⁶. Additionally, other reports suggest that polythiophene's orientation is enhanced by iodine-covered Au(111)⁴⁰. These phenomena are attributed to epitaxy, where polymers grow along the crystalline lattice on the substrate due to interactions between halogen atoms and polymers. In this study, the alignment of GNRs is expected to be improved by epitaxy based on π - π interactions between GNRs on the top layer and short chain of GNRs in the underlayer.”

References

- “40. Sakaguchi, H., Matsumura, H. & Gong, H. Electrochemical epitaxial polymerization of single-molecular wires. *Nat. Mater.* **3**, 551–557 (2004).
46. Abyazisani, M., MacLeod, J. M. & Lipton-Duffin, J. Cleaning up after the Party: Removing the Byproducts of On-Surface Ullmann Coupling. *ACS Nano* **13**, 9270–9278 (2019).”

Q10: Why was an iodine-covered Au(111) substrate instead of a bare Au(111) chosen for the investigation of multilayer-growth of 5-AGNRs?

A10: Sakaguchi *et al.* (*Nat. Mater.* **3**, 551–557 2004) synthesized polythiophene on a gold substrate and observed polymer chains using STM. They also found that iodine-covered Au(111) substrates improved lattice matching between polythiophenes and substrates, resulting in improvement of polymer orientations. Our study found that electrochemically produced GNRs could be synthesized on both Au(111) and iodine-covered Au(111) substrates. However, we observed that the orientation of GNRs improved when synthesized on an iodine-covered Au(111) substrate. This improvement is due to lattice matching between GNRs and the substrate.

We added the following sentences to the text and modified Supplementary Fig. 2.

Page 4, Line 4-10

“Atomic flat Au(111) and iodine-covered Au(111) substrates were utilized for scanning tunneling microscopy (STM) measurements. Iodine-covered Au(111) improves the alignment of GNRs due to lattice matching between GNRs and the iodine atoms on the substrate (Supplementary Figs. 1 and 2). A transparent ITO substrate was used for optical measurements. The reaction temperature of 80°C enhances formation of GNRs by overcoming activation energy and improving precursor diffusion, compared to room temperature (Supplementary Fig. 3)^{41,42}.”

Page 4, Line 26-31

“Lattice parameters of iodine-covered Au(111) in a compressed hexagonal structure, along the a, b and c axes are 4.2, 5.4 and 4.2 Å, respectively, while those of Au(111) are 2.9 Å. The periodicity of GNR (8.5 Å) mismatches with the atom spacing of substrates

(Au(111) and iodine-covered Au(111)) by 3.53% and 1.17%, respectively. This indicates that iodine-covered Au(111) is more suitable than Au(111) for epitaxial GNR growth.”

Supplementary Fig. 2. (a) LT-STM image of iodine-covered Au (111) with a superimposed structure of GNR. The inset shows a, b, and c axes, and a compressed hexagonal lattice structure. I-I distances along the a, b, and c axes are 4.2, 5.4, and 4.2 Å, respectively. The mismatch between GNR and substrate is 1.17%. (b) Lattice matching between the GNR and Au(111). Au-Au distances along the a, b, and c axes are 2.9, 2.9, 2.9 Å, respectively. The mismatch between GNR and substrate is 3.53%.

Q11: The applications of electrochemically produced GNRs should be included in the conclusion section.

A11: We added this to the Conclusions.

Page 13, Line 22-25

“Our electrochemical on-surface synthesis can be applied to various aromatic precursors, leading to wider GNRs. Moreover, electrochemically produced GNRs present excellent silicon-etching properties compared to noble metals, implying great potential for Si nanolithography and exhibit excellent photoconductivity.”

Reviewers' Comments:

Reviewer #1:

Remarks to the Author:

In the revision, the authors have fully addressed my concerns. I would suggest to publish this nice piece of GNR work as it is.

Reviewer #2:

Remarks to the Author:

I appreciate the effort (including additional experiments) undertaken by the authors to improve the manuscript and address raised questions.

I believe that the authors addressed the most important issues in a satisfactory manner and therefore I recommend the revised manuscript for publication in Nature Communications.

Reviewer #3:

Remarks to the Author:

Most of my previous comments are satisfactorily addressed, thus I recommend its publication in Nat. Commun. and further review is not needed. Please consider the following minor comments to further improve the manuscript.

1. "This minor difference between experimental results and DFT calculations, may be attributed to the excluded intermolecular interactions". The influence of intermolecular interactions on the bandgap is usually not large. Other reasons include electron screening by a metal substrate and charge transfer, as well as the precision of DFT.
2. "The differential conductance map was recorded with set sample set at -0.2 V or 0.6 V and I at 80 pA by a digital lock-in amplifier with a modulation frequency of 350 Hz and an AC voltage of 2 mV." JUST A SUGGESTION for your future works: the parameters of 80 pA and 2 mV are too small for dI/dV maps (2 mV is generally used for the mapping between -100 mV 100 mV or even smaller). 800 pA and 20 mV would be better to get a good resolution.

Response to Reviewer 3

Q1 represents the reviewer's comment on the bandgap difference between experimental and theoretical values, while Q2 is a suggestion for experimental dI/dV mapping measurements. We aim to address Q1.

Q1: "This minor difference between experimental results and DFT calculations, may be attributed to the excluded intermolecular interactions". The influence of intermolecular interactions on the bandgap is usually not large. Other reasons include electron screening by a metal substrate and charge transfer, as well as the precision of DFT.

A1: We modified the phrase.

Page 4, Line 9-11

“This minor difference between experimental results and DFT calculations may be attributed to electron screening by a metal substrate and charge transfer between GNRs and the substrate.”